# Hydration product phase evolution and mortar strength development in alkali-activated slag and fly ash systems

Zhuo Jin[1], Aimin Gong [1]*, Yier Huang[1], Yulin Peng[2], Kang Yong[1], Shanqing Shao[1]

1 College of Water Conservancy, Yunnan Agricultural University, Kunming, China, 2 Academic Affairs Office, Yunnan Agricultural University, Kunming, China

* 13708457658@163.com

## Abstract

Alkali-activated geopolymer materials, derived predominantly from industrial byproducts such as fly ash and slag, represent a sustainable alternative to Portland cement for applications including anti-seepage grouting, road construction, and high-strength concrete. This study systematically investigates the hydration behavior of slag and fly ash activated by NaOH and $Ca(OH)_2$ at dosages of 4%, 6%, and 8%, with the constraint that the initial setting time is $\geq 45$ min and the final setting time is $\leq 600$ min. The mechanical properties of the resultant mortar systems were evaluated using standardized strength testing (ISO method) at curing ages of 3, 7, and 28 days. The phase composition and microstructural evolution of hydration products were characterized using scanning electron microscopy (SEM), X-ray diffraction (XRD), Fourier transform infrared spectroscopy (FTIR), backscattered electron image analysis (BSE-IA), and isothermal calorimetry. These analytical techniques provided comprehensive insights into the morphology, phase distribution, porosity, and hydration kinetics of the reaction products. The results revealed distinct activator-dependent reactivity trends: NaOH demonstrated higher efficiency in activating slag, whereas $Ca(OH)_2$ was more effective in promoting the hydration of fly ash. Optimal hydration was achieved with 8% NaOH for slag and 6% $Ca(OH)_2$ for fly ash, leading to enhanced reaction completeness, increased hydration product formation, denser pore structures, and significantly improved mechanical properties. Alkali-activated slag exhibited substantially greater strength enhancement than fly ash. The 28-day compressive strengths reached 35.94 MPa and 6.65 MPa for slag- and fly ash-based mortars, respectively, with corresponding flexural strengths of 10.23 MPa and 1.92 MPa. These findings demonstrate that the properties of alkali-activated geopolymer materials can be effectively tailored through the strategic selection of alkaline activator type and dosage. This study provides both theoretical insights and technical guidance for the development of sustainable alkali-activated geopolymer materials in construction applications.

**Data availability statement:** All data files are available from the figshare database (DOI: 10.6084/m9.figshare.29365352).

**Funding:** The author(s) received no specific funding for this work.

**Competing interests:** The authors have declared that no competing interests exist.

## 1. Introduction

Cement stands as one of the most extensively consumed construction materials worldwide. However, its conventional production process is associated with substantial carbon emissions and high energy consumption [1,2]. These challenges not only impose severe environmental impacts but also hinder the green transformation and sustainable development of the construction industry [3]. In this context, geopolymers have emerged as a promising sustainable alternative. First proposed by French scientist Joseph Davidovits in 1978, this class of materials utilizes alkaline activators to dissolve and polycondense industrial solid wastes (e.g., slag, fly ash) or natural aluminosilicates (e.g., metakaolin), which possess pozzolanic or latent hydraulic activity, to form a solid cementitious matrix [4–6]. In comparison to ordinary Portland cement (OPC), alkali-activated geopolymer binders offer a distinct advantage: they can be synthesized without the energy-intensive high-temperature clinker calcination step, leading to a significant reduction in carbon emissions [7,8]. As cementitious materials, they exhibit a suite of desirable properties, including low heat of hydration, excellent binding capacity, and superior long-term durability. Notably, these materials can maintain their structural integrity and performance over extended periods, even under complex and aggressive environmental conditions [9–11]. Owing to this combination of environmental and technical benefits, geopolymers demonstrate considerable application potential across a diverse range of fields. These include conventional construction and transportation infrastructure, as well as specialized domains such as hydraulic and marine engineering, structural repair and strengthening, 3D printing of building components, and environmental remediation applications like the immobilization of heavy metal contaminants.

Granulated blast furnace slag (GBFS), a by-product of ironmaking [12], is primarily composed of $CaO$, $SiO_2$, and $Al_2O_3$. Fly ash (FA), generated during coal-fired power generation [13], consists mainly of $SiO_2$ and $Al_2O_3$. This chemical composition imparts significant pozzolanic activity to both GBFS and FA, making them valuable supplementary cementitious materials [14,15]. Despite their beneficial properties, the current comprehensive utilization rate of these industrial byproducts remains suboptimal. Therefore, a systematic investigation of their performance in alkali-activated systems is imperative. Such research is crucial for facilitating the large-scale valorization of these solid wastes, mitigating the environmental footprint associated with their stockpiling, and establishing a robust scientific foundation for their targeted engineering applications. Among the multitude of factors influencing the properties of alkali-activated cementitious materials, the type and dosage of the alkaline activator are recognized as being of paramount importance [16]. Consequently, numerous researchers have extensively studied the performance of GBFS and FA under alkaline activation. For instance, Sun et al. [17] and Li et al.[18] demonstrated that both GBFS and FA exhibit high reactivity under alkaline activation, creating favorable conditions for the development of mechanical properties; however, the type and concentration of the alkaline activator significantly influence their mechanical performance. Liu et al.[19] found that NaOH effectively activated FA, achieving a compressive strength of 4.21 MPa in cement-fly ash matrices, indicating NaOH's

distinct advantage in enhancing FA reactivity and improving the mechanical properties of cementitious materials. Qiao et al.[20] reported that, compared with $Na_2SiO_3$, NaOH resulted in a denser microstructure and significantly higher strength in cement-fly ash-based cemented matrices, with the most pronounced improvement observed in the 28-day compressive strength. Ge et al. [21] observed that the compressive strength of FA-based matrices activated by a mixed solution of NaOH and $Na_2SiO_3$ increased continuously throughout the curing period, albeit at a gradually decreasing rate. Han et al. [22] showed that partially replacing cement with GBFS improved the microstructure of cement mortar, reducing microcracks and pores, enhancing the matrix's splitting tensile strength, and improving the ductility and deformability of cement concrete. This suggests good compatibility of GBFS in cementitious systems and its ability to enhance mechanical properties through microstructural refinement. Yu et al. [23] investigated the effects of NaOH and sodium silicate (water glass) on the compressive strength and chloride ion erosion resistance of slag concrete. Their results indicated that specimens activated by sodium silicate exhibited higher compressive strength than those activated by NaOH, whereas NaOH-activated specimens demonstrated superior resistance to chloride ion erosion. Zuo et al.[24] demonstrated that the reaction degree of slag gradually increased as the NaOH content increased from 4% to 8%.

In summary, current research on alkali-activated geopolymer materials has primarily focused on the strength development trends of specimens activated by single or compound alkaline activators, with relatively limited investigation into the evolution of hydration product phases. Given that the type and dosage of activators significantly influence the performance of alkali-activated cementitious systems, this study investigates the phase evolution of hydration products in slag/fly ash-based systems under varying dosages of NaOH and $Ca(OH)_2$. The work specifically aims to optimize alkaline activator dosages to enhance the mechanical properties of alkali-activated geopolymer binders, thereby providing a theoretical foundation for the resource utilization of these industrial by-products.

## 2 Experimental materials and methods

### 2.1 Raw materials

The raw materials used in this study are shown in Fig 1.

Granulated Blast Furnace Slag (GBFS): S95-grade GBFS (conforming to Chinese standard GB/T 18046) was supplied by Gongyi Longze Water Purification Material Co., Ltd. Its physical properties included a Blaine specific surface area of 429 m²/kg, a density of 2.8 g/cm³, and a particle size distribution predominantly in the range of 1–100 μm. Its chemical composition meets the S95 standard requirements, with a fluidity ratio ≥ 95%, a 7-day activity index ≥ 75%, and a 28-day activity index ≥ 95%. Detailed chemical compositions are presented in Table 1.

Fly Ash (FA): Class F fly ash (according to ASTM C618) was obtained from Gongyi Run Refractories Co., Ltd. It has a density of 2.1 g/cm³ and a particle size distribution similar to that of the slag (1–100 μm). Detailed chemical compositions are presented in Table 1.

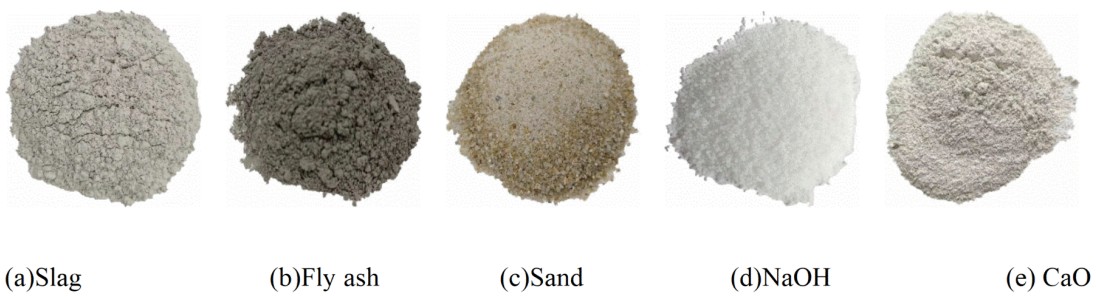

| (a)Slag | (b)Fly ash | (c)Sand | (d)NaOH | (e) CaO |

**Fig 1. Raw materials.**

**Table 1. Chemical composition of fly ash and slag.**

| Constituent | Content (wt%) | | | | | | | | |
|---|---|---|---|---|---|---|---|---|---|
| | CaO | $SiO_2$ | $Al_2O_3$ | $SO_3$ | $Fe_2O_3$ | MgO | $Na_2O$ | $K_2O$ | other |
| Slag | 34 | 34.5 | 17.7 | 1.64 | 1.03 | 6.01 | — | — | 5.12 |
| Fly ash | 4.5 | 45.1 | 36.8 | 1.2 | 0.85 | — | — | — | 11.55 |

Alkaline activators: Sodium Hydroxide (NaOH): Solid pellets with a purity ≥ 98% were sourced from Tianjin Zhiyuan Chemical Reagent Co., Ltd. An 8 mol/L concentrated solution was prepared by dissolving the pellets in deionized water 24 hours before use to dissipate the heat of dissolution. The pH of the prepared solution was 14.3.

Calcium Hydroxide ($Ca(OH)_2$): This activator was prepared by slaking "Hui Hui" brand quicklime (CaO) powder with deionized water. Chemical titration confirmed that the resulting slurry contained 93.45% $Ca(OH)_2$ by mass. The slurry was also prepared 24 hours in advance and had a pH of 12.4.

Fine Aggregate: ISO standard sand conforming to the specification of ISO 679 was used.

Mixing Water: Deionized water was used for all mixtures and solution preparations.

## 2.2 Mix proportions and specimen preparation

**2.2.1 Mix proportions.** To eliminate the influence of heat released from the alkaline activators on the test results, the NaOH solutions and $Ca(OH)_2$ slurries were prepared 24 hours in advance and stored in sealed containers at room temperature (23 ± 2°C). The mix proportions for all tested pastes and mortars are detailed in Table 2.

**2.2.2 Specimen fabrication and testing.** 2.2.2.1 Determination of setting time: The initial and final setting times of the cement pastes were determined using a Vicat apparatus (ISO standard) following the Chinese National Standard GB/T 1346–2011, which is analogous to ASTM C191. The water demand for normal consistency was first determined for each mixture. The cement paste was then prepared using 500 g of cementitious material (slag or fly ash) and the corresponding amount of water (including both the mixing water and the water introduced with the activators). The mixing procedure consisted of an initial 120 s of low-speed mixing (140 ± 5 rpm), a 15 s pause for scraping down the bowl, followed by 120 s of high-speed mixing

**Table 2. Mix proportions.**

| Sample | Material | Alkaline activator type | Activator Dosage (wt%) |
|---|---|---|---|
| 4SN | Slag | NaOH | 4 |
| 6SN | Slag | NaOH | 6 |
| 8SN | Slag | NaOH | 8 |
| 4SC | Slag | $Ca(OH)_2$ | 4 |
| 6SC | Slag | $Ca(OH)_2$ | 6 |
| 8SC | Slag | $Ca(OH)_2$ | 8 |
| 4FN | Fly ash | NaOH | 4 |
| 6FN | Fly ash | NaOH | 6 |
| 8FN | Fly ash | NaOH | 8 |
| 4FC | Fly ash | $Ca(OH)_2$ | 4 |
| 6FC | Fly ash | $Ca(OH)_2$ | 6 |
| 8FC | Fly ash | $Ca(OH)_2$ | 8 |

**Note:** The numbers 4, 6, and 8 denote the dosage of alkaline activators as a percentage of the total mass of cementitious materials. S represents slag, F represents fly ash, N denotes NaOH, C denotes $Ca(OH)_2$. For example, 6SN indicates the addition of 6% NaOH to slag, and other notations follow this convention.

(285±10 rpm). The mixed paste was immediately cast into the Vicat ring mold and placed in a moist curing cabinet (20±1°C, relative humidity≥90%). Measurements commenced 30 minutes after adding water. The initial setting time was defined as the time when the needle penetrated to a depth of 4±1 mm from the bottom of the mold. The final setting time was recorded when the needle made an indentation not exceeding 0.5 mm in diameter on the paste surface. Only mixtures meeting the practical criteria of an initial setting time≥45 min and a final setting time≤600 min were selected for subsequent mortar testing.

2.2.2.2 Mortar strength testing: Mortar strength was determined according to GB/T 17671−2021 (Method for Testing Cementitious Sand Strength – ISO Method) for slag-based and fly ash-based systems. Based on Table 2 proportions, constituent materials were mixed in a cement mortar mixer, then cast into 40 mm×40 mm×160 mm prism molds using a ZS-15 ISO-compliant mortar vibrator. Nine specimens were prepared per mixture. After 24-hour ambient curing (23±2°C), specimens were demolded and transferred to a standard curing chamber (20±2°C, RH≥95%) until testing ages (3d, 7d, 28d). Flexural and compressive strengths were determined using an automated testing machine. Specimen fabrication and testing equipment are illustrated in Fig 2.

2.2.2.3 Microstructural Analysis (SEM, XRD, FTIR, BSE-IA): Following the compressive strength tests, fragments from the crushed specimens were immediately immersed in absolute ethanol for at least 72 h to terminate hydration. The samples were subsequently dried in a vacuum oven at 60°C for 24 h and then prepared for the following microstructural analyses:

Scanning Electron Microscopy (SEM): Dried specimen fragments with fresh fracture surfaces were mounted on aluminum stubs, sputter-coated with a thin layer of gold to enhance conductivity, and examined using a Hitachi S-3400N scanning electron microscope operating under high vacuum conditions.

X-ray Diffraction (XRD): The dried fragments were ground into a fine powder using an agate mortar and pestle and passed through a 75 µm sieve. XRD analysis was performed using a MiniFlex600 diffractometer (Rigaku) with Cu-Kα

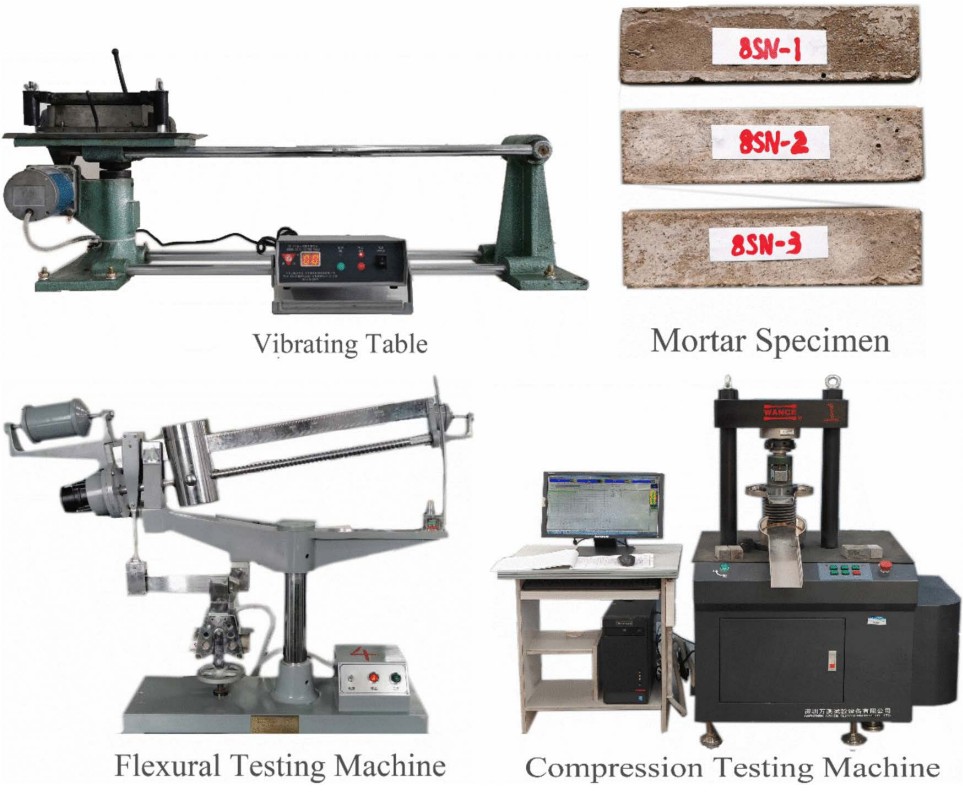

**Fig 2. Mortar specimen and testing equipment.**

radiation ($\lambda = 1.5406$ Å). Diffraction patterns were collected over a 2θ range of 10° to 90° with a step size of 0.02° and a scanning speed of 2°/min.

Fourier Transform Infrared Spectroscopy (FTIR): The same powdered samples prepared for XRD analysis were used for FTIR characterization. Spectra were acquired using a Nicolet iS50 FTIR spectrometer (Thermo Scientific) via the KBr pellet method, scanning a wavenumber range from 525 to 4000 cm$^{-1}$ with a spectral resolution of 4 cm$^{-1}$.

Backscattered Electron Image Analysis (BSE-IA): This technique was used to characterize the pore structures of the 28-day cured mortar specimens. The acquired BSE images underwent grayscale calibration and median filtering for noise reduction. A thresholding procedure was then applied to segment the pores (dark regions) from the solid phases, enabling quantitative porosity analysis.

2.2.2.4 Isothermal calorimetry: The hydration heat release of the paste samples was measured using an eight-channel TAM Air isothermal calorimeter (TA Instruments, USA) maintained at a constant temperature of 20.0 ± 0.1°C. Approximately 5 g of freshly mixed paste was promptly transferred into a sealed glass ampoule and placed in the calorimeter. The heat flow rate and cumulative heat evolution were monitored continuously for 72 h. Data analysis was performed using the instrument's proprietary software.

## 3 Results and discussion

### 3.1 Setting time

The setting time is a critical parameter governing both the practical workability and the development of mechanical properties of cementitious materials. The influence of NaOH and Ca(OH)$_2$ on the setting behavior of the slag-based and fly ash-based systems is shown in Figs 3 and 4, respectively. A comparison of these figures reveals that the setting times of the slag-based system were substantially shorter than those of the fly ash-based system across all tested activator dosages. This indicates more rapid initial hydration reaction kinetics in the slag-based system under alkaline activation. Furthermore, a distinct activator-dependent trend was observed: NaOH markedly accelerated the setting of the slag-based system compared to Ca(OH)$_2$, whereas it considerably prolonged the setting of the fly ash-based system. These results demonstrate the higher sensitivity of slag to NaOH activation and the preferential response of fly ash to Ca(OH)$_2$ activation. This

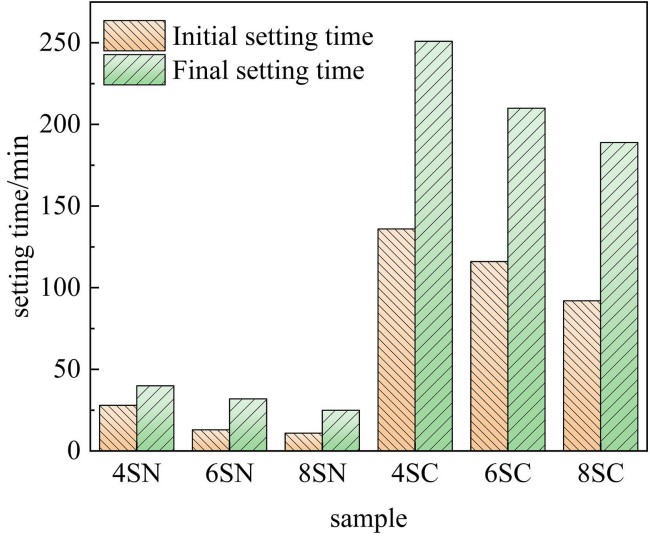

**Fig 3. Setting time of slag.**

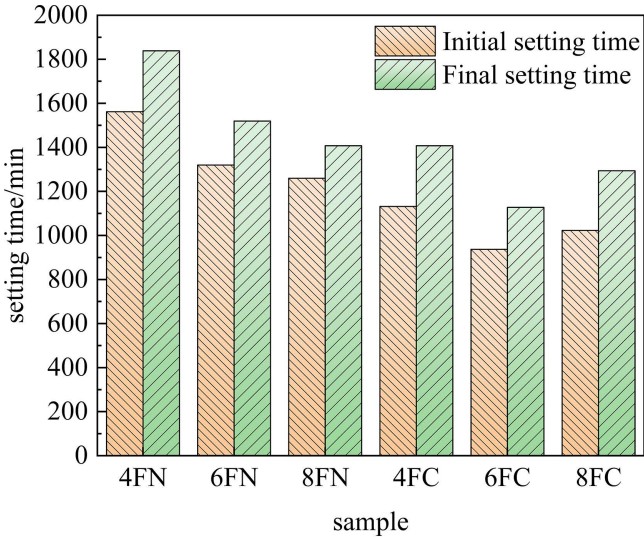

**Fig 4. Setting time of fly ash.**

divergent behavior is primarily attributable to the role of the Ca/Si ratio. In the high-calcium slag-based matrix, the introduction of additional $Ca^{2+}$ via $Ca(OH)_2$ further elevates the Ca/Si ratio. An excessively high Ca/Si ratio can inhibit the dissolution of key reactive phases from the slag particles, thereby retarding the setting process. In contrast, for the low-calcium fly ash system, which is inherently calcium-deficient, the increased Ca/Si ratio provided by $Ca(OH)_2$ facilitates the precipitation of strength-giving hydration products such as C-(A)-S-H gel, thereby accelerating the setting reaction. The observed dependence of setting behavior on the Ca/Si ratio is consistent with findings reported by Zhao et al. [25].

As is evident from Fig 3, increasing the dosage of either NaOH or $Ca(OH)_2$ from 4% to 8% led to a progressive reduction in the setting times of the slag-based system. This trend demonstrates that the enhanced alkalinity provided by both activators promotes the dissolution of reactive silicate and aluminate species from the slag, consequently accelerating the nucleation and growth of hydration products and leading to faster setting and hardening. As shown in Fig 4, a similar trend was observed for the fly ash system activated by NaOH, where higher dosages (4% to 8%) resulted in progressively shorter setting times. Conversely, for $Ca(OH)_2$ activation, the setting time initially decreased but then increased as the dosage was raised from 4% to 8%, indicating a non-linear, optimal dosage effect. This confirms that the increased alkalinity from NaOH effectively depolymerizes the glassy aluminosilicate phases in fly ash, enhancing its reactivity. A moderate increase in $Ca(OH)_2$ dosage provides a favorable alkaline environment (high pH) while supplying necessary $Ca^{2+}$ ions, synergistically enhancing fly ash dissolution and subsequent gel formation. However, an excessive $Ca(OH)_2$ dosage (e.g., 8%) appears to oversaturate the system with $Ca^{2+}$ and $OH^-$ ions, which can lead to the premature precipitation of less-reactive phases or passivation of the fly ash particle surfaces, thereby retarding the overall reaction and prolonging the setting time. In summary, the setting behavior of these alkali-activated systems is highly tunable through the strategic selection and dosage of the alkaline activator, allowing for customization to meet specific application requirements.

### 3.2 Mortar strength

Compressive and flexural strength are fundamental mechanical properties of cementitious materials, serving as key indicators for structural integrity and long-term durability. The development of compressive and flexural strength in the slag-based system, as influenced by the type and dosage of NaOH and $Ca(OH)_2$, is shown in Figs 5 and 6, respectively.

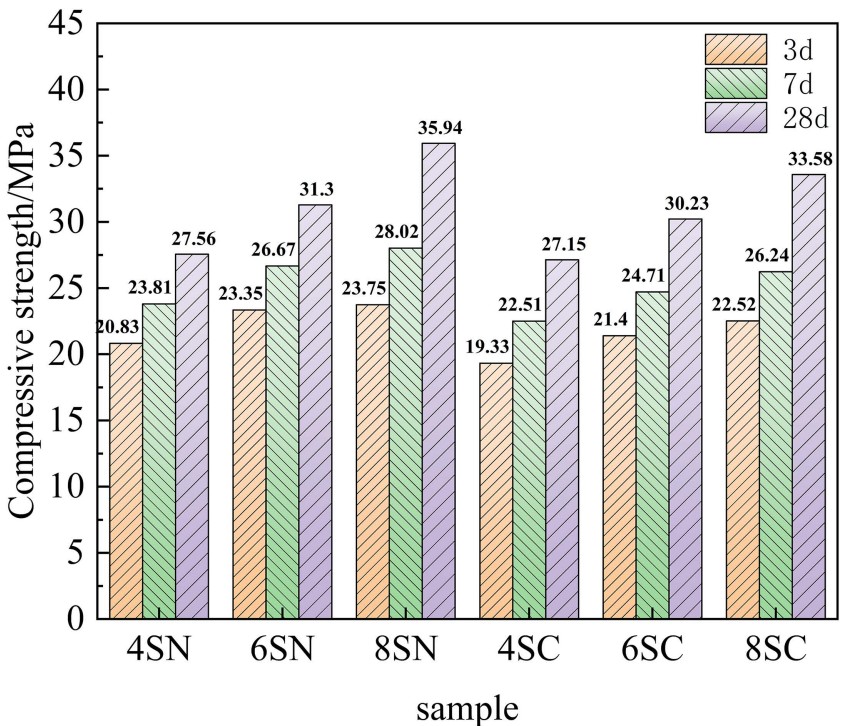

**Fig 5. Compressive strength of slag.**

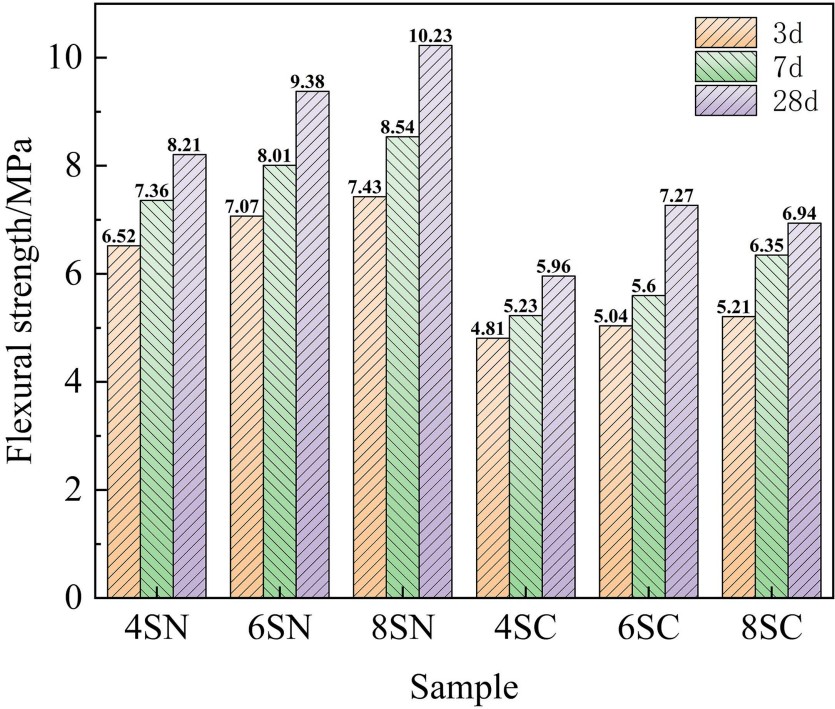

**Fig 6. Flexural strength of slag.**

A clear superiority of NaOH over Ca(OH)$_2$ as an activator for slag is evident from these figures, with NaOH yielding significantly higher compressive and flexural strengths across all curing ages and dosage levels. This performance disparity can be ascribed to more rapid hydration kinetics and the formation of a greater volume of effective binding hydration products (e.g., C-(A)-S-H gel) in the NaOH-activated systems, confirming the superior reactivity of slag under highly alkaline conditions provided by NaOH. Furthermore, a positive correlation was observed between the NaOH dosage (from 4% to 8%) and the mechanical strength development in the slag-based system. This enhancement is directly attributable to the elevated alkalinity, which promotes the depolymerization of the vitreous slag structure by breaking Si-O-Si and Si-O-Al bonds. This process liberates a higher concentration of silicate ($[SiO_4]^{4-}$) and aluminate ($[AlO_4]^{5-}$) monomers. These species subsequently react with Ca$^{2+}$ (from the slag) and Na$^+$ (from the activator) in the pore solution, precipitating a cohesive matrix primarily composed of C-(A)-S-H gel, with potential co-formation of N-A-S-H type gels, which collectively contribute to strength development. The intergrowth and space-filling nature of these gels significantly refine the pore structure, reducing porosity and enhancing the overall density of the matrix. This is the fundamental mechanism behind the strength improvement, an interpretation consistent with the mechanistic insights provided by Zhong et al.[26] and Li et al. [27]. In contrast, the effect of Ca(OH)$_2$ on strength development was less straightforward. While compressive strength showed gradual improvement with increasing Ca(OH)$_2$ dosage from 4% to 8%, the flexural strength displayed a more complex trend: it increased gradually at 3 and 7 days, but showed an initial rise followed by a decline at 28 days for the higher dosages.

The compressive and flexural strength development of the fly ash-based system, activated by NaOH and Ca(OH)$_2$, is shown in Figs 7 and 8, respectively. The results clearly indicate that Ca(OH)$_2$ activation yields superior mortar strength compared to NaOH activation for the fly ash system, demonstrating that Ca(OH)$_2$ is a more effective activator for unlocking the pozzolanic potential of fly ash. This enhancement is primarily attributed to the crucial role of calcium. The addition of Ca(OH)$_2$

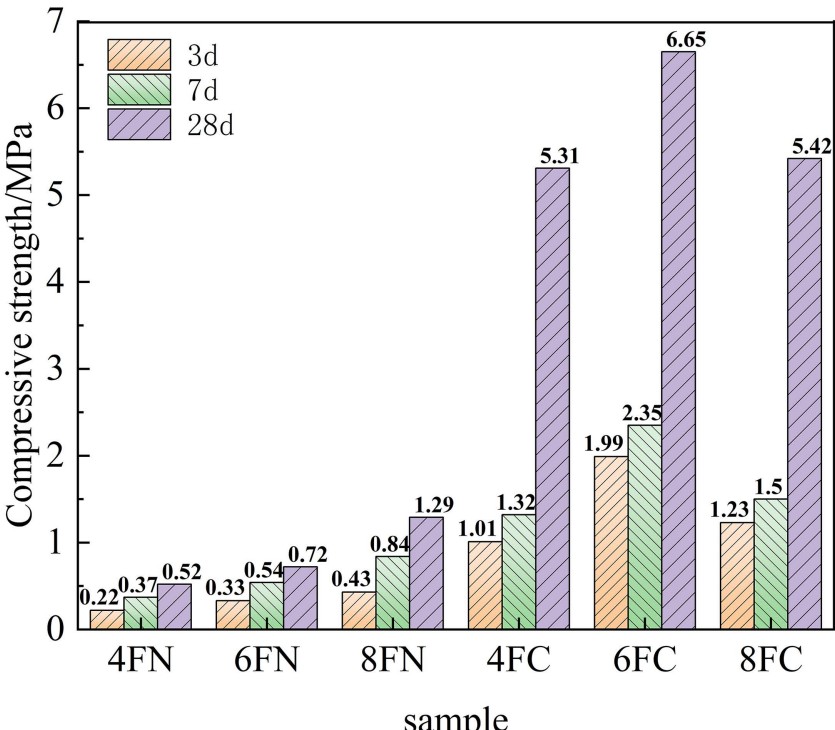

**Fig 7. Compressive strength of fly ash.**

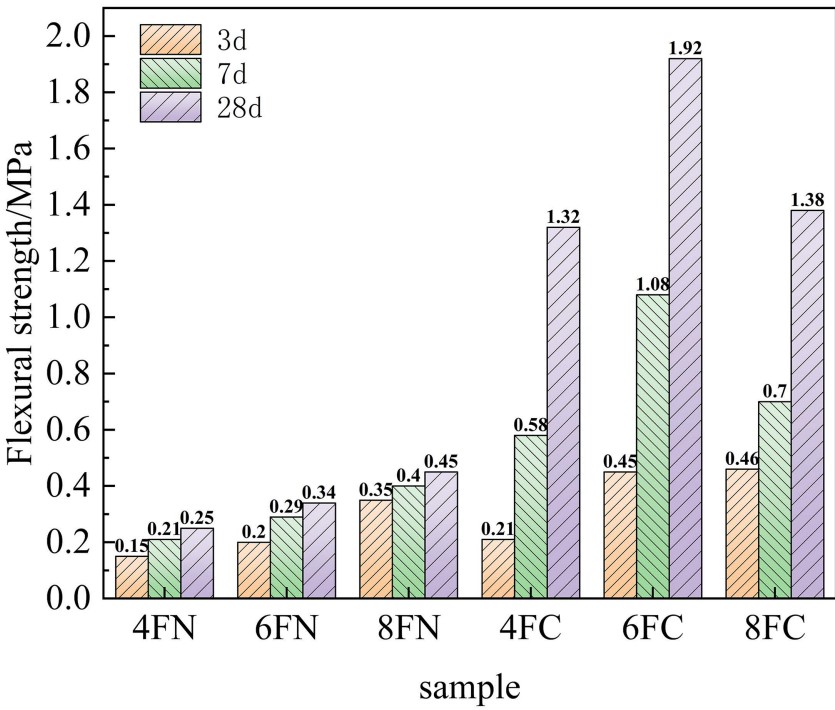

**Fig 8. Flexural strength of fly ash.**

increases the Ca/Si ratio in the system, which is essential for the formation of strength-giving calcium (alumino)silicate hydrate (C-(A)-S-H) gels, thereby accelerating the reaction kinetics and strength development. Furthermore, the introduced $Ca^{2+}$ ions can participate in the reaction pathways, potentially leading to the formation of more stable and denser C-(A)-S-H gel, either by modifying the N-A-S-H gel structure or by forming independently. The coexistence and intergrowth of these gel phases create a more cohesive and denser microstructure, which substantially enhances the mechanical strength. Increasing the NaOH dosage from 4% to 8% led to a progressive, though modest, improvement in strength, confirming that higher alkalinity facilitates the dissolution of fly ash particles. Conversely, for $Ca(OH)_2$ activation, an optimal dosage is observed. The strength increases as the dosage rises from 4% to 6%, but a further increase to 8% results in a noticeable decrease in both compressive and flexural strength. This strength reduction at the 8% $Ca(OH)_2$ dosage is attributed to the dual effect of excessive alkalinity and an overly high Ca/Si ratio. This unfavorable condition may lead to the rapid precipitation of less effective or porous reaction products, potentially passivating the remaining fly ash particles and hindering further depolymerization of the glassy phase and the formation of a cohesive gel matrix. The effectiveness of $Ca(OH)_2$, despite its limited solubility, stems from its ability to establish a sufficiently high pH environment upon dissolution. The released $OH^-$ ions initiate the depolymerization of the fly ash glassy phase by breaking Si-O-Si and Si-O-Al bonds, liberating reactive silicate and aluminate species. Concurrently, the introduced $Ca^{2+}$ ions readily combine with these species, promoting the nucleation and growth of C-(A)-S-H gel. The precipitation of this gel lowers the ion concentration in the solution, thereby maintaining a chemical driving force for the ongoing dissolution of fly ash via Le Chatelier's principle, which underpins the continuous strength development.

### 3.3 SEM analysis

Scanning Electron Microscopy (SEM) was used to establish the relationship between the macroscopic mechanical properties and the underlying microstructure of the alkali-activated slag and fly ash systems, providing visual evidence of hydration product morphology and matrix development.

Ca(OH)$_2$-activated Slag: In the system activated with 6% Ca(OH)$_2$ (6SC), the early hydration stage (3 days, Fig 9a) is characterized by the formation of abundant but structurally loose and porous C-(A)-S-H gel with a long-chain morphology, only partially covering the unhydrated slag particles. This underdeveloped microstructure correlates with the lower early-age strength of this mixture. By 28 days (Fig 9c), these products underwent dehydration and polycondensation, forming a relatively continuous microstructure, although significant large pores remained. In contrast, the system with 8% Ca(OH)$_2$ (8SC) produced substantial flocculent C-(A)-S-H gel and acicular ettringite (AFt) crystals at 3 days (Fig 9d), rather than a continuous network. The quantity of AFt increased at 7 days (Fig 9e), which is attributed to the elevated Ca²⁺concentration and enhanced alkalinity accelerating dissolution and promoting AFt formation. The expansive nature of AFt mitigated shrinkage deformation, explaining the higher strength of the 8SC group compared to the 6SC group. By 28 days (Fig 9f), a substantial increase in hydration products effectively filled the interstitial spaces, resulting in a more uniform, dense, and continuous binding matrix.

NaOH-activated Slag: The system with 6% NaOH (6SN) showed a significant decrease in unhydrated slag particles at 3 days (Fig 9g), forming loosely structured flocculent C-(A)-S-H gel intergrown with platy gismondine at 7 days (Fig 9h). The interpenetrating C-(A)-S-H gel and gismondine contributed to strength development, although numerous large pores persisted. By 28 days (Fig 9i), the volume of hydration products increased, rod-like structures disappeared, and a compact microstructure developed. Compared to 6SN, the 8% NaOH system (8SN) exhibited a higher degree of hydration product polymerization at 3 days (Fig 9j) and formed a relatively dense microstructure by 7 days (Fig 9k). Further dehydration and polycondensation at 28 days (Fig 9l) yielded an even denser structure, with the 8SN sample exhibiting the most compact microstructure among all samples.

Collectively, the SEM analysis confirms the superior effectiveness of NaOH over Ca(OH)$_2$ in activating slag, resulting in a denser microstructure. The 8% NaOH dosage (8SN) facilitated the most complete reaction and the formation of the most compact matrix, which aligns perfectly with its superior mechanical performance and is consistent with the findings of Zuo et al.[24]

The microstructural development of the fly ash-based system, corresponding to the strength results, is shown in Fig 10, illustrating the distinct effects of Ca(OH)$_2$ and NaOH activation over time.

Ca(OH)$_2$-activated Fly Ash: At early ages (3 and 7 days), the microstructure was dominated by a substantial quantity of unreacted and agglomerated fly ash particles (Fig 10a, 10b, 10d, 10e). Only limited formation of hydration products was observed at this stage, appearing as porous, flocculent, or reticulated gels, tentatively identified as C-(A)-S-H and N-A-S-H. These initial gels primarily adhered to particle surfaces or partially filled interparticle spaces, indicating an ongoing but incomplete reaction during the first week. By 28 days (Fig 10c, 10f), a significant progression is evident. A considerably larger volume of hydration products formed, effectively enveloping the residual fly ash particles and binding them together into a more continuous and polymerized matrix. However, the persistent presence of discernible, partially reacted particles underscores the slower reaction kinetics of fly ash compared to slag. Notably, the sample with the optimal 6% Ca(OH)$_2$ dosage (6FC, Fig 10c) developed the densest and most homogeneous microstructure among the Ca(OH)$_2$-activated groups, with fewer visible pores and a more extensive gel network, providing a microstructural explanation for its peak mechanical performance.

NaOH-activated Fly Ash: In stark contrast, the microstructures of the NaOH-activated systems (Fig 10g-10l) are characterized by the persistent and abundant presence of unhydrated fly ash particles throughout the entire 28-day curing period. This visual evidence strongly indicates the low activation efficiency of NaOH alone for this Class F fly ash, as it failed to significantly depolymerize the glassy phases. Only isolated and limited formations of flocculent hydration products are observed, which failed to develop into a pervasive, continuous binding matrix. This lack of effective microstructural cohesion directly explains the poor strength development in these mixtures.

## 3.4 XRD analysis

The degree of hydration reaction is a critical indicator of the hardening process in cementitious materials. To assess the evolution of hydration products and precursor consumption under the influence of different alkaline activators and

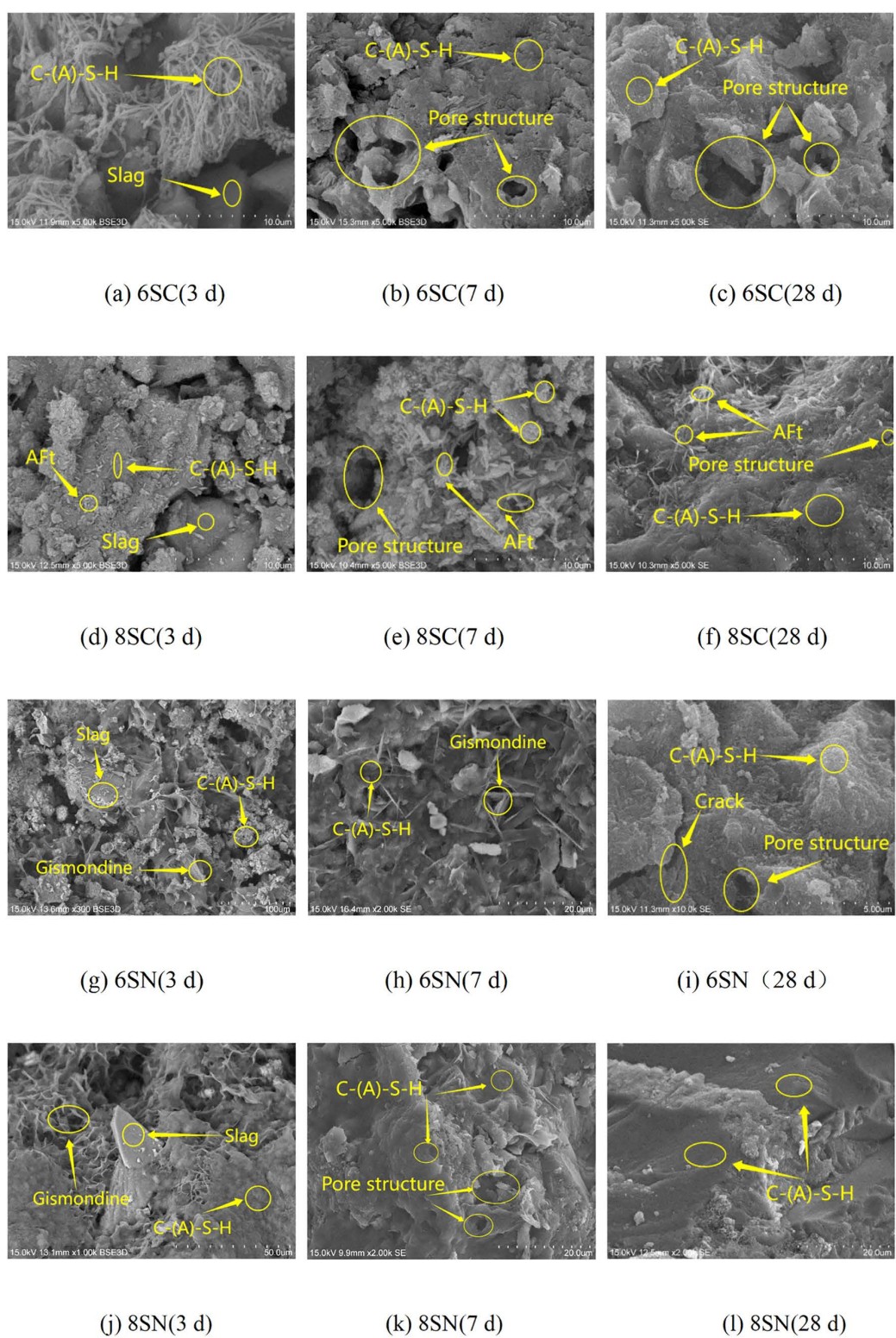

(a) 6SC(3 d)     (b) 6SC(7 d)     (c) 6SC(28 d)

(d) 8SC(3 d)     (e) 8SC(7 d)     (f) 8SC(28 d)

(g) 6SN（3 d）     (h) 6SN(7 d)     (i) 6SN（28 d）

(j) 8SN(3 d)     (k) 8SN(7 d)     (l) 8SN(28 d)

**Fig 9. SEM images of slag activated by 6% and 8% Ca(OH)₂ or NaOH.**

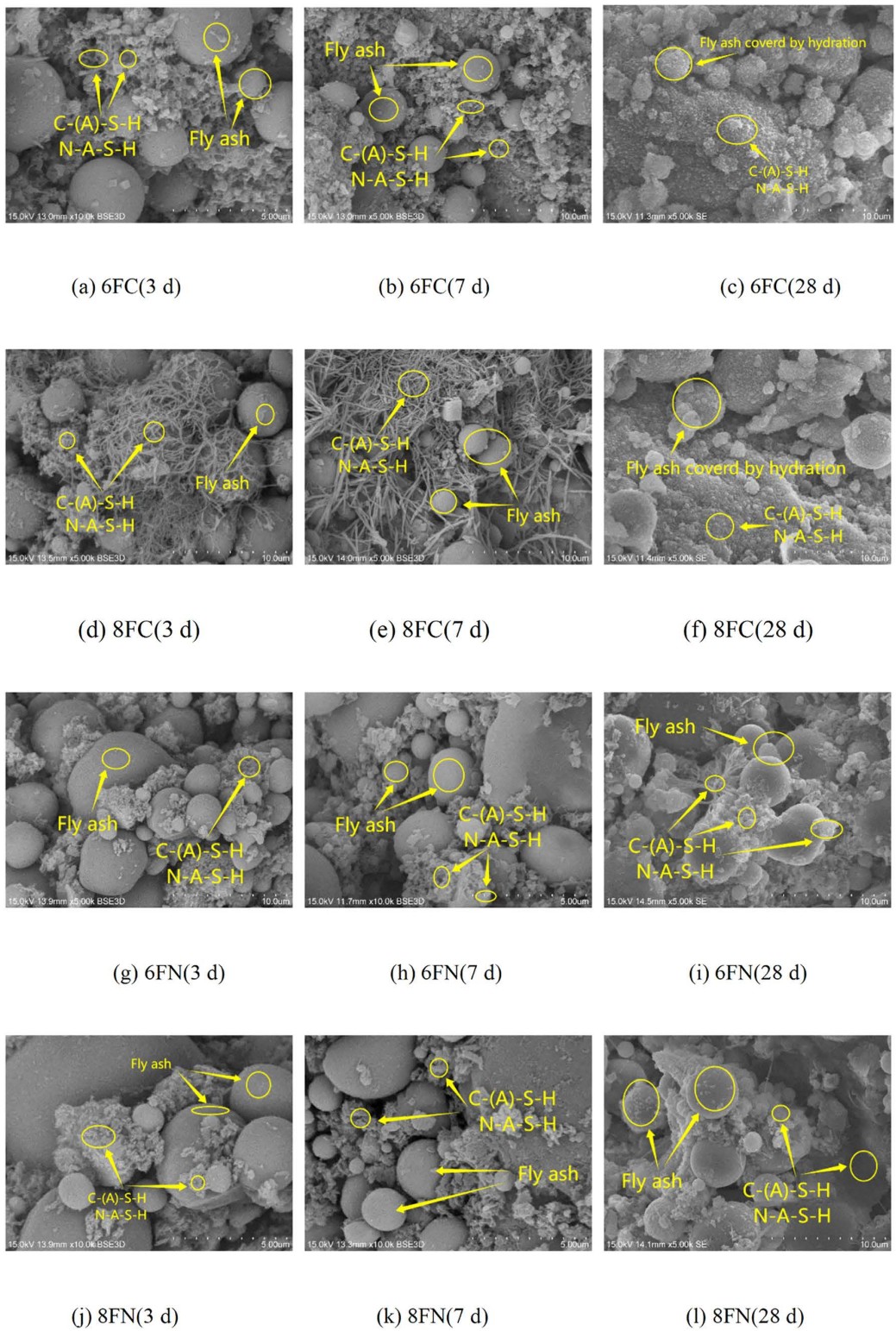

(a) 6FC(3 d)  (b) 6FC(7 d)  (c) 6FC(28 d)

(d) 8FC(3 d)  (e) 8FC(7 d)  (f) 8FC(28 d)

(g) 6FN(3 d)  (h) 6FN(7 d)  (i) 6FN(28 d)

(j) 8FN(3 d)  (k) 8FN(7 d)  (l) 8FN(28 d)

**Fig 10. SEM images of fly ash activated by 6% and 8% NaOH or Ca(OH)$_2$.**

dosages, X-ray diffraction (XRD) analysis was performed on specimens cured for 28 days. A semi-quantitative assessment based on the relative intensities of characteristic diffraction peaks was conducted to evaluate the influence of alkaline activation on the reaction degree, as shown in Figs 11 and 12.

As expected, the primary hydration products—amorphous C-(A)-S-H and N-A-S-H gels—do not yield distinct diffraction peaks due to their disordered atomic structure and thus manifest as a broad hump in the 2θ range of approximately 10°–90°. In the slag-based system (Fig 11), the identified crystalline phases primarily consist of unreacted quartz ($SiO_2$) from the raw slag and gismondine ($CaAl_2Si_2O_8 \cdot 4H_2O$, a zeolite-like phase), which is a crystalline hydration product. In the fly ash-based system (Fig 12), the dominant crystalline phases are unreacted quartz ($SiO_2$) and mullite ($3Al_2O_3 \cdot 2SiO_2$)

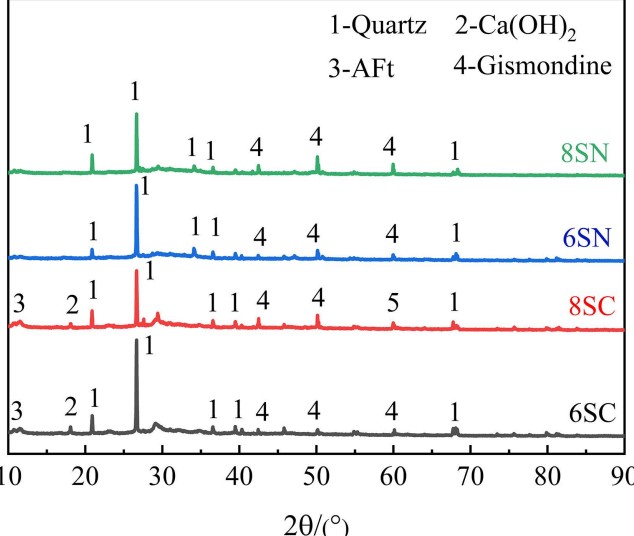

**Fig 11. XRD patterns of the slag-based system.**

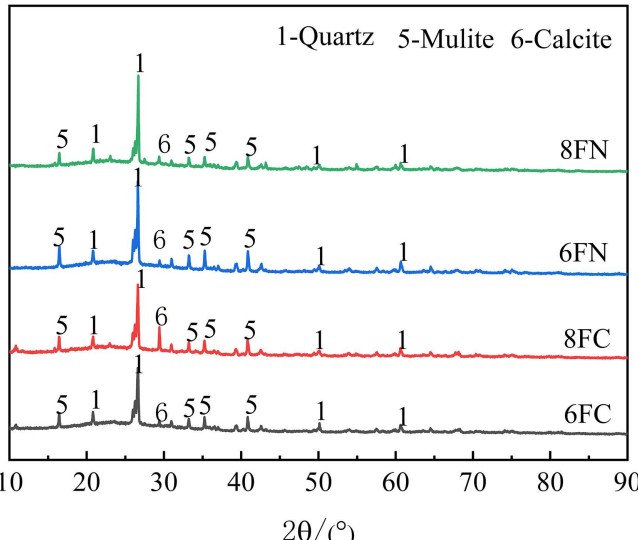

**Fig 12. XRD patterns of the fly ash-based system.**

from the fly ash precursor. Calcite ($CaCO_3$) was also detected, indicating partial carbonation, likely of portlandite or calcium-containing gels. The persistence of quartz and mullite peaks indicates unreacted particles, while a reduction in their intensity signifies consumption during reaction to form amorphous gels. Similarly, in slag systems, decreased quartz intensity implies a higher degree of reaction. The detection of gismondine confirms crystalline hydration product formation under alkaline conditions. The significant differences in relative peak intensities among mixtures demonstrate that the type and dosage of the alkaline activator profoundly influence reaction kinetics and the ultimate degree of precursor conversion.

In the slag-based system (Fig 11), sample 8SN exhibited a lower quartz peak intensity and a higher gismondine peak intensity compared to 6SN. This indicates that the higher 8% NaOH dosage promoted more extensive dissolution of the slag, resulting in a greater volume of both amorphous gels (contributing to the broad hump) and crystalline gismondine. A similar, though less pronounced, trend was observed for $Ca(OH)_2$ activation, where 8SC showed reduced quartz intensity relative to 6SC, confirming that a higher $Ca(OH)_2$ dosage also facilitates slag reaction. The lowest quartz intensity and the most intense gismondine peaks were consistently observed in the 8SN group. This confirms that an 8% NaOH dosage provides the most effective activation conditions for dissolving the glassy phase of slag, thereby maximizing the formation of both the amorphous binding gels and the associated crystalline phases. Portlandite ($Ca(OH)_2$) peaks are detectable in both 8SC and 6SC patterns, indicating either an excess of the activator beyond what could be consumed in the reaction or its formation from free CaO in the slag. Additionally, a minor ettringite (AFt) peak was identified at approximately 11.44° 2θ, which originates from the reaction between dissolved $Ca^{2+}$, aluminate species, and sulfates inherently present in the slag. In the fly ash-based system (Fig 12), a pivotal finding is the significantly lower quartz and mullite peak intensities in the $Ca(OH)_2$-activated specimens (6FC, 8FC) compared to their NaOH-activated counterparts (6FN, 8FN). This provides direct mineralogical evidence of $Ca(OH)_2$'s superior efficacy in dissolving the aluminosilicate phases of fly ash, thereby facilitating the formation of amorphous C-(A)-S-H and N-A-S-H gels. This finding is consistent with the mechanism proposed by Zhao et al.[25] for $Ca(OH)_2$-activated systems. Among the NaOH-activated specimens, 8FN exhibited reduced quartz and mullite intensities relative to 6FN, confirming that increased alkalinity does enhance the dissolution of fly ash and the formation of N-A-S-H gel, albeit to a much lesser extent than $Ca(OH)_2$ activation. Conversely, and in alignment with the strength results, the 8FC specimen exhibited higher quartz and mullite peak intensities than the optimal 6FC specimen. This indicates that the excessive 8% $Ca(OH)_2$ dosage inhibits the dissolution of fly ash particles. Consequently, the formation of strength-contributing amorphous gels is impaired, providing a clear phase-compositional explanation for the observed strength deterioration at this high dosage. The weakest intensities for the crystalline fly ash phases (quartz, mullite) occurred in the 6FC specimen, identifying it as the formulation with the highest degree of fly ash consumption and thus confirming 6% as the optimal $Ca(OH)_2$ dosage for activating this specific fly ash. The presence of calcite ($CaCO_3$) peaks confirms some carbonation occurred in both 8FC and 6FC. The notably stronger calcite peak in 8FC is logically attributed to the carbonation of a larger amount of unreacted portlandite, which is consistent with its higher initial dosage and its inferred inhibitory effect on the hydration reaction, as suggested by the higher residual quartz/mullite content.

### 3.5 FTIR analysis

Since the mechanical strength of these materials is primarily derived from the amorphous gel phases, which lack distinct diffraction patterns in XRD, Fourier-transform infrared (FTIR) spectroscopy was employed to probe the molecular structure of the hydration products. This technique probes the vibrational modes of chemical bonds. Analysis of the resulting spectra allows for a semi-quantitative assessment of gel formation and provides insights into the aluminosilicate network's degree of polymerization (DOP), thereby helping to elucidate the strength differences between the systems.

The FTIR spectra of specimens cured for 28 days are shown in Figs 13 and 14. The key absorption bands for analysis are in the region of 1300–850 cm$^{-1}$, corresponding to the asymmetric stretching vibrations of Si–O–T bonds (where T

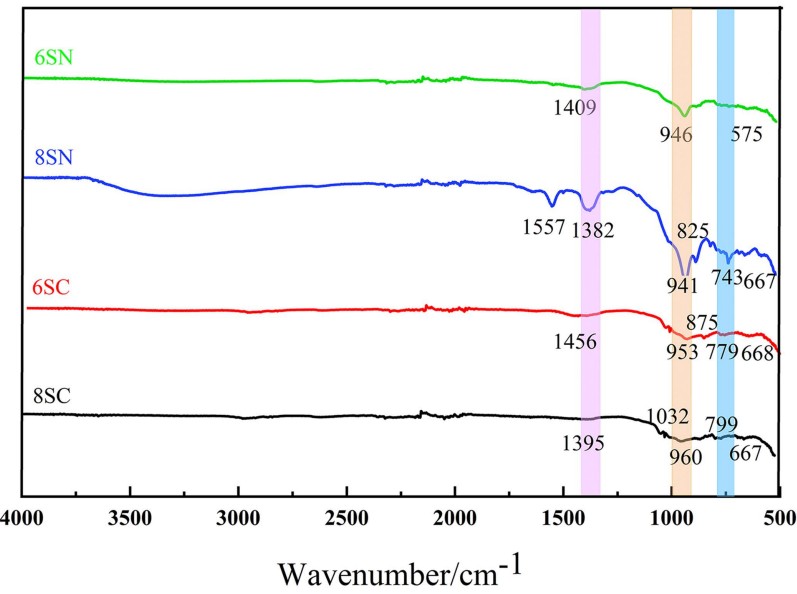

**Fig 13. FTIR spectra of the slag-based system.**

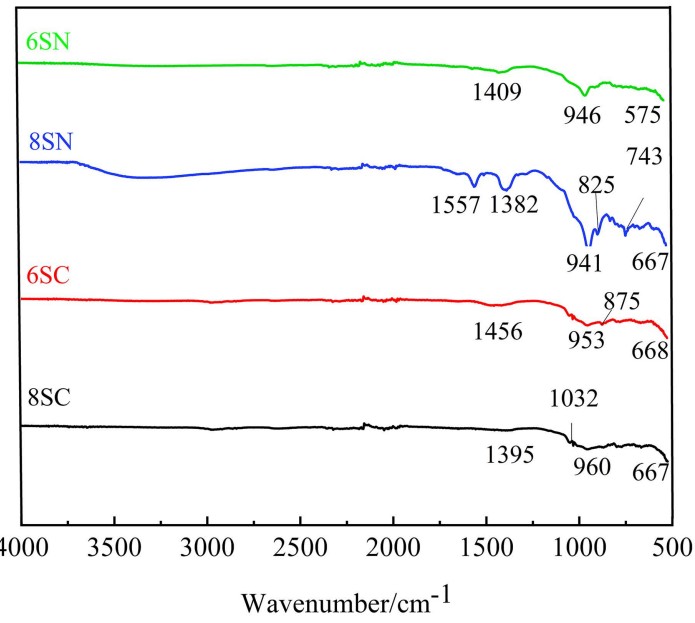

**Fig 14. FTIR spectra of the fly ash-based system.**

is Si or Al). The position and shape of this main band are sensitive to the DOP and the Si/Al ratio. Additionally, the peak near 743 cm$^{-1}$ is attributed to symmetric stretching vibrations of Si–O–Si bonds, further confirming the presence of an amorphous aluminosilicate gel network. All spectra exhibit absorption bands in these characteristic regions, indicating

that the fundamental building blocks of the reaction products ($SiO_4$ and $AlO_4$ tetrahedra) are consistent across all mixtures, primarily forming amorphous aluminosilicate gels. This observation aligns with the findings of Tao et al.[28].

Slag-based System (Fig 13): The main Si–O–T absorption band is significantly more intense for the NaOH-activated slag groups (8SN, 6SN) than for the $Ca(OH)_2$-activated groups (8SC, 6SC), suggesting a greater quantity of gel phases and providing molecular-level evidence for NaOH's superior efficacy. Within the NaOH-activated series, the 8SN specimen exhibits a more intense Si–O–T band than 6SN, indicating a higher volume of reaction products. A key observation is the downward shift of the main Si–O–T band (e.g., from ~946 $cm^{-1}$ in 6SN to ~941 $cm^{-1}$ in 8SN), typically indicating a higher degree of Al substitution for Si and/or a slight decrease in DOP. The substantial increase in total gel volume in 8SN appears to outweigh potential nanostructural changes, resulting in superior mechanical performance dominated by gel quantity. For $Ca(OH)_2$-activated slag, the 6SC specimen shows a more intense Si–O–T band than 8SC, located at a higher wavenumber. A shift to a higher wavenumber can be associated with a higher proportion of $Q^3$ units relative to $Q^2$ units, suggesting a gel with a higher DOP and potentially improved structural stability, aligning with the findings of Wan et al.[29].

Fly Ash-based System (Fig 14): The Si–O–T band intensity is stronger in 8FN than in 6FN, whereas the opposite is true for $Ca(OH)_2$-activated systems (6FC > 8FC). This demonstrates that increased NaOH dosage promotes fly ash dissolution and N-A-S-H gel formation, while the higher 8% $Ca(OH)_2$ dosage inhibits depolymerization, resulting in less gel formation, consistent with XRD and strength data. Furthermore, $Ca(OH)_2$-activated systems (6FC, 8FC) generally display more intense Si–O–T bands at higher wavenumbers compared to NaOH-activated systems (6FN, 8FN). This indicates that not only is the quantity of gels greater with $Ca(OH)_2$ activation, but the gels also possess a more polymerized structure. This molecular-level evidence conclusively confirms the pronounced sensitivity and superior reactivity of this fly ash to $Ca(OH)_2$ activation, providing a fundamental explanation for the superior strength development and shorter setting times.

### 3.6 Pore phase distribution and pore structure

The pore structure, including its volume, size distribution, and spatial arrangement, is a critical microstructural characteristic that governs key macroscopic properties such as strength, durability, and frost resistance. In this study, the pore characteristics of the 28-day cured mortar specimens were quantitatively analyzed using Backscattered Electron Image Analysis (BSE-IA).

NaOH-activated Slag: Visual comparison of the BSE images (Fig 15a, 15b) indicates a notably higher pore density and more pronounced pore clustering in the 6SN specimen compared to 8SN. The quantitative data in Table 3 corroborate this: the 8SN mixture exhibits substantially lower total porosity, a smaller maximum pore diameter, a reduced total number of pores, and a finer pore size distribution. This indicates that the 8 wt.% NaOH dosage induces a more complete reaction, generating a greater volume of hydration products that effectively fill and segment the capillary pore space, thereby refining the pore structure, a mechanism consistent with the findings of Li et al. [30].

$Ca(OH)_2$-activated Slag: Similarly, the 8SC specimen (Fig 15d) exhibits a lower number of pores and a more uniform spatial distribution compared to 6SC (Fig 15c). Quantitatively, the total porosity of 8SC is 14.56% lower than that of 6SC, accompanied by a reduction in total pore count and a significant decrease in maximum pore diameter. This demonstrates that the higher 8 wt.% $Ca(OH)_2$ dosage is more effective in modifying the reaction products and resulting microstructure, leading to a denser matrix. A comparative analysis confirms that the 8SN specimen possesses the most favorable pore structure among the slag-based mixtures, characterized by the lowest total porosity, the virtual absence of large detrimental pores, and the most homogeneous pore distribution.

Fly Ash Systems: For the fly ash systems, analysis of the $Ca(OH)_2$-activated specimens (Fig 15e, 15f) indicates that the optimal 6FC mixture possesses a lower pore density and less pronounced pore clustering compared to the 8FC mixture. The data in Table 3 quantitatively support this, showing that 6FC has lower porosity, fewer pores, a smaller maximum pore diameter, and a finer pore system overall. This confirms that the 6 wt.% $Ca(OH)_2$ dosage promotes a more complete reaction of fly ash, leading to an optimized internal microstructure with a refined pore system. Such a pore structure is known

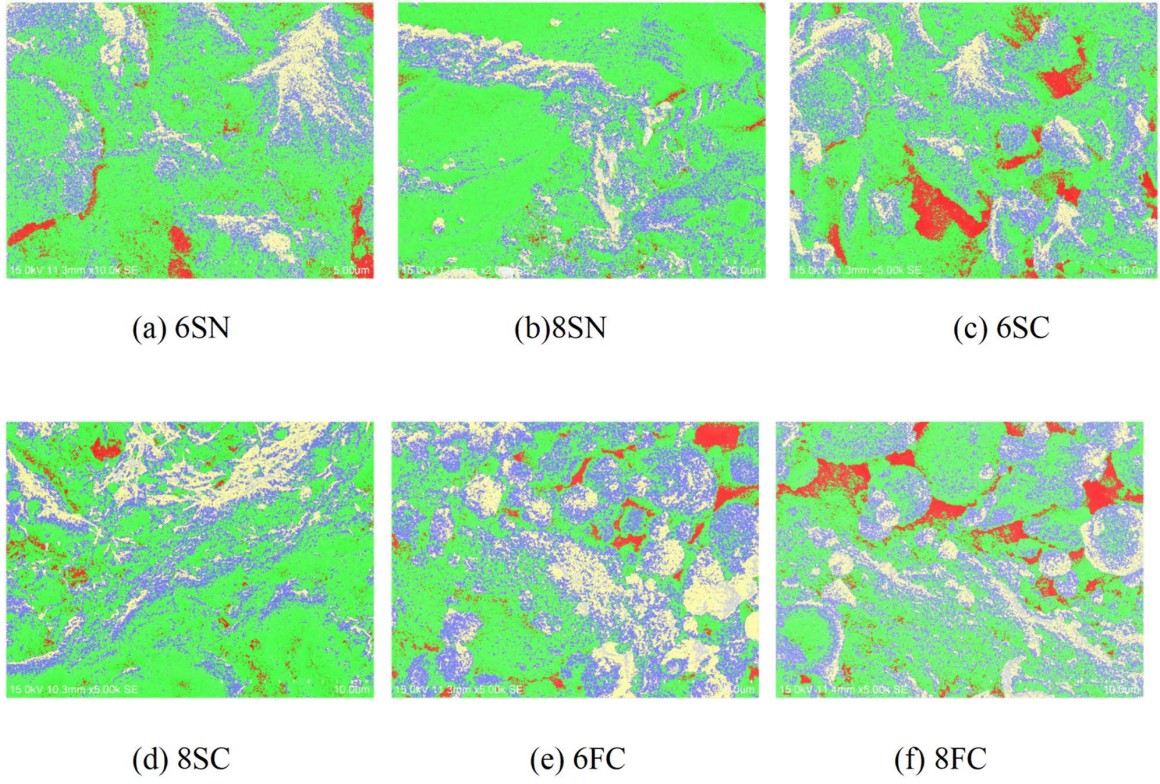

(a) 6SN　　　　　　(b)8SN　　　　　　(c) 6SC

(d) 8SC　　　　　　(e) 6FC　　　　　　(f) 8FC

**Fig 15. Pore phase distribution maps(BSE-IA) of the slag and fly ash-based systems.**

**Table 3. Characteristic pore structure parameters.**

| sample | Porosity/% | Number of pores per unit/ piece | Maximum pore diameter/μm | Total pore area/mm² |
|---|---|---|---|---|
| 6SN | 35.83 | 4456 | 2.325 | 0.0105 |
| 8SN | 25.12 | 3598 | 0.206 | 0.0572 |
| 6SC | 46.06 | 4140 | 3.028 | 0.0194 |
| 8SC | 31.50 | 2712 | 0.726 | 0.0158 |
| 6FC | 45.59 | 3618 | 2.854 | 0.0217 |
| 8FC | 48.68 | 5685 | 3.624 | 0.0192 |

to impede water ingress and ice formation, thereby enhancing durability factors like frost resistance, which aligns with the principles discussed by Bernal et al. [11].

## 3.7 Hydration Characteristics Analysis

To gain deeper insight into the hydration reaction kinetics, the heat evolution of the optimally performing mixtures—slag activated with 8 wt.% NaOH (8SN) and fly ash activated with 6 wt.% $Ca(OH)_2$ (6FC)—was monitored using an isothermal calorimeter over 72 hours. Figs 16 and 17 show the corresponding hydration heat release rate and cumulative heat release curves, respectively.

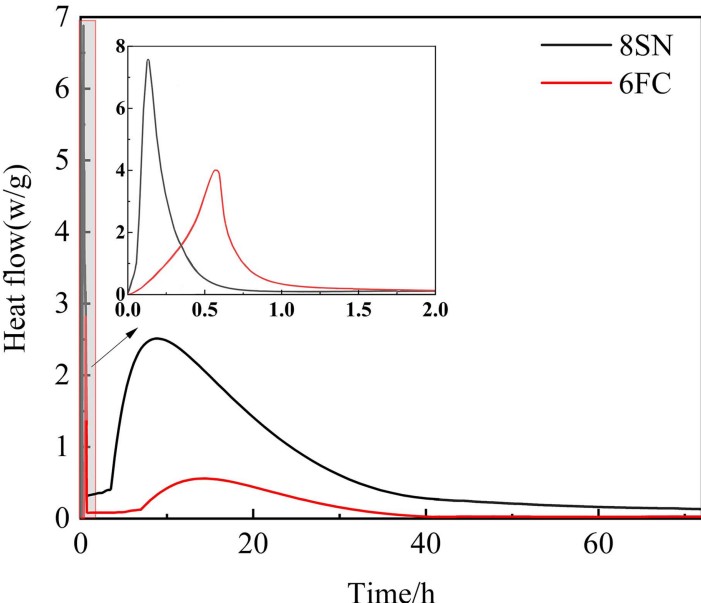

**Fig 16. Hydration heat release rate.**

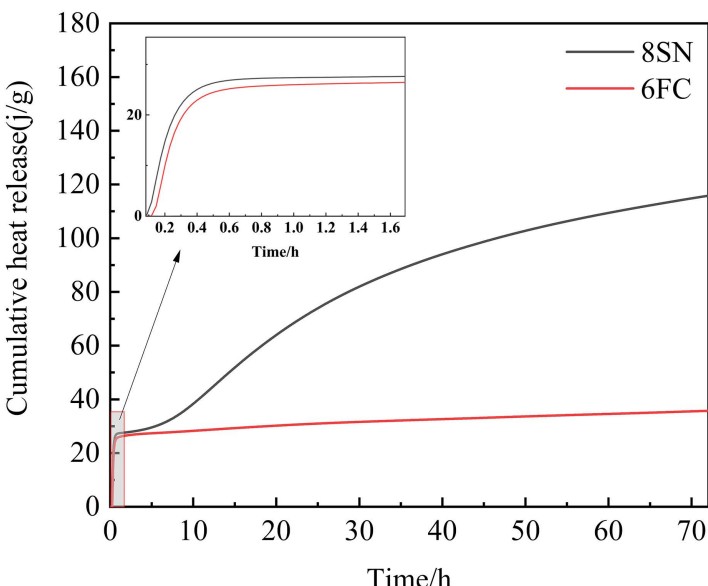

**Fig 17. Cumulative hydration heat.**

Slag System (8SN): As shown in Fig 16, the slag system exhibits a sharp, intense exothermic peak immediately after mixing (within ~0.2 h). This peak is attributed to initial wetting, ion dissolution, and rapid formation of early reaction products, correlating with its very short setting time and early-age strength development. A second, broader exothermic peak follows, associated with the main reaction period involving accelerated nucleation and growth of the C-(A)-S-H gel phase.

The cumulative heat release curve for slag (Fig 17) shows a high total heat output that increases rapidly during the first 24 h and continues to rise steadily thereafter, signifying a high overall reaction extent and substantial formation of binding gels responsible for microstructural densification. The calorimetry data thus establish a characteristic three-stage reaction regime for the optimally activated slag: (i) an immediate, intense reaction governing initial set and early strength; (ii) a main reaction period driving primary strength gain; and (iii) a sustained later reaction contributing to continued microstructural refinement.

Fly Ash System (6FC): In stark contrast, the fly ash system exhibits a much less intense initial exothermic peak, which occurs later and is broader, centered around 0.6 h (Fig 16). This signifies a slower and less vigorous initial dissolution and reaction process. The subsequent heat release rate remains at a low but sustained level over several days, without a distinct, sharp second peak, suggesting a slow, continuous reaction mechanism characteristic of the gradual dissolution of fly ash and precipitation of gels. Consequently, the cumulative heat release for fly ash (Fig 17) is significantly lower and increases more gradually than that of slag. This low heat evolution profile is a hallmark of alkali-activated fly ash systems, reflecting their longer setting times and slower strength development kinetics. The sustained reaction, however, facilitates the development of a relatively uniform and dense microstructure at later ages.

### 3.8 Technical limitations

Despite the insightful findings of this work, several limitations should be acknowledged, which also delineate productive avenues for subsequent research.

Firstly, the setting behavior was assessed using the standard Vicat needle method. Its reliance on manual operation introduces a potential for subjective error. More importantly, it offers only discrete measurements and cannot capture the continuous, dynamic evolution of the early-age hydration and structural build-up processes. To overcome this limitation, future investigations should employ automated monitoring techniques, such as the sensor-based data acquisition approaches referenced in [31,32]. This would yield a more objective, high-resolution dataset of the setting kinetics, enabling a deeper understanding of the early-stage reaction mechanisms.

Secondly, the present study focused on the fundamental properties of setting time and mechanical strength under standard curing conditions. For the successful transition of these alkali-activated systems from laboratory research to real-world engineering applications, two critical aspects demand further exploration: (1) the long-term durability under aggressive environments, and (2) the development of reliable models for predicting long-term performance. Future work will, therefore, utilize methodologies akin to those in [33] to establish a robust predictive model for strength development over time. Concurrently, following the approaches detailed in [34–36], a comprehensive evaluation of durability—focusing on resistance to sulfate attack, chloride penetration, and freeze-thaw cycles—will be undertaken. Addressing these aspects is imperative to build confidence and provide the essential technical foundation for the broad adoption of these sustainable binders.

### Conclusion

This study presented a systematic investigation into the influence of alkaline activator type (NaOH and $Ca(OH)_2$) and dosage (4%, 6%, and 8%) on the setting behavior, mechanical strength development, hydration product evolution, and microstructure of alkali-activated slag and fly ash systems. The key conclusions are summarized as follows:

(1) The setting kinetics were strongly dependent on both the precursor and activator type. For slag systems activated by either NaOH or $Ca(OH)_2$, and for fly ash systems activated by NaOH, increasing the activator dosage from 4% to 8% consistently accelerated the setting and hardening process. In contrast, for the $Ca(OH)_2$-activated fly ash system, a non-monotonic trend was observed, where the setting time reached a minimum at 6% dosage before increasing at 8%, indicating an optimal dosage beyond which retardation occurs.

(2) The mechanical performance was directly linked to the efficacy of the activator-precursor combination. The slag system exhibited superior sensitivity to NaOH activation, achieving its highest 28-day compressive strength (35.94 MPa) and the densest microstructure at an 8% dosage. Conversely, the fly ash system was most effectively activated by $Ca(OH)_2$, reaching its peak compressive strength (6.65 MPa) at an optimal dosage of 6%.

(3) Microstructural and calorimetric analyses revealed distinct reaction mechanisms. The optimally activated slag system (8% NaOH) exhibited a vigorous, multi-stage hydration process, leading to the rapid formation of a dense matrix. The optimally activated fly ash system (6% $Ca(OH)_2$), while reacting more gradually, facilitated the steady formation of a homogeneous and cohesive microstructure, supporting sustained strength development.

(4) Quantitative pore structure analysis confirmed that the formulations delivering the highest mechanical strength—namely, the slag system with 8% NaOH and the fly ash system with 6% $Ca(OH)_2$—were also characterized by the most refined microstructures, exhibiting the lowest total porosity, the smallest maximum pore diameters, and a uniform pore distribution.

## Author contributions

**Conceptualization:** Zhuo Jin.

**Data curation:** Zhuo Jin, Yier Huang, Yulin Peng.

**Formal analysis:** Kang Yong, Shanqing Shao.

**Methodology:** Zhuo Jin, Yier Huang, Yulin Peng.

**Supervision:** Aimin Gong, Shanqing Shao.

**Writing – original draft:** Zhuo Jin.

**Writing – review & editing:** Aimin Gong.

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
