## [Decision Letter · Decision Letter 0]

7 May 2025

Dear Dr. Gong,

Thank you for submitting your manuscript to PLOS ONE. After careful consideration, we feel that it has merit but does not fully meet PLOS ONE’s publication criteria as it currently stands. Therefore, we invite you to submit a revised version of the manuscript that addresses the points raised during the review process.

We look forward to receiving your revised manuscript.

Kind regards,

Parthiban Kathirvel

Academic Editor

PLOS ONE

4. Please amend your authorship list in your manuscript file to include author 镯 金.

Reviewers' comments:

Reviewer's Responses to Questions

**Comments to the Author**

1. Is the manuscript technically sound, and do the data support the conclusions?

Reviewer #1: No

Reviewer #2: No

Reviewer #3: Partly

2. Has the statistical analysis been performed appropriately and rigorously?

Reviewer #1: No

Reviewer #2: No

Reviewer #3: N/A

3. Have the authors made all data underlying the findings in their manuscript fully available?

Reviewer #1: No

Reviewer #2: Yes

Reviewer #3: Yes

4. Is the manuscript presented in an intelligible fashion and written in standard English?

Reviewer #1: No

Reviewer #2: No

Reviewer #3: No

Reviewer #1: The submission overall follows the lab instruction and is straightforward. It mimics the Chinese guidelines and thus based on other international known guidelines like ASTM its generalizing might become difficult. Overall, it fails to highlight gaps or biases in the literature, and provides uncharacterized research gaps leading to unjustified conclusions without meaningful insights or future directions.

Most of the references are in Chinese which are not applicable for international readers

Structurally disorganized. It doesn’t follow the IMRAD. It has figs with different parts while almost none of them either in the context or in the captions have been mentioned or discussed. For example, Fig 7, 15 …

Sensitivity analysis as described in this work is DECLINED to be accepted

Discussion on XRD

Lack of convincing and documented Discussion: technical limitations/uncertainty analysis/feasibility study and possibility for large scale use/benefits to industry/solid comparison with other works to show the improvement/the impact of bias of the used data on the results/…

Substandard English/unreflective and general keywords/…

When we discuss setting time, procedure requires continuous updating (dynamic databases) while this work doesn’t provide a solution for uniformly normalizing dynamic databases via different factors. Search for new approaches via keywords like normalizing large scale sensor-based data with an automated method …

In terms of the strength (Fig 5, 6…) you may benefit from other works like https://link.springer.com/article/10.1007/s41062-016-0016-9...

Conclusion should be justified

Reference list should follow the journal guidelines and supplemented by DOI link

Reviewer #2: Title

The title should be rephrased, fly ash and slag should appear in the title

Abstract

The abstract lacks quantity, include results obtained for compressive strength and flexural strength. Ensure that you contrast between results obtained for slag and fly ash.

Rewrite the abstract to improve readability

Correct misspelt words

Introduction

The justification for the study is not well established. Improve on this

Rephrase sentence: is China's major contributor to emissions

Use fullstop (.) before The

Start as a new paragraph

Methods

Improve this paragraph to aid coherency. Also provide pictures of the materials used. It is important to provide the particle size distribution of the slag and the fly ash.

Provide detailed information about this procedures. Just as was done for compressive strength. Describe all the testing methods to a great detail in different subsections. XRD, FTIR, TAM, BSE

Results

What do you mean by change rule diagram

The sentence is too long. Consider breaking the sentences to improve readability.

Improve readability. Discuss relative to existing literature

The method for sensitivity analysis was not captured in the methods

Highlight the important minerals and show the observable changes using inscribed rectangles.

FTIR analysis not FIRT Analysis

Introduce a shape here to call the reader to the difference in intensity.

Improve the discussion by making reference to recent literature. (Hydration characterization)

Describe the method in detail under the method section. (Pore physical phase distribution and pore structure characteristics)

Consider rephrasing the sentence. Also improve this section with information from literature. What is the ideal pore area for geopolymer composites based on previous attempts by other researchers?

Conclusion

Provide a brief information about the purpose of the study before itemising the conclusions. Also, include the limitations of this study after the conclusion. Furthermore, I will suggest you give the entire manuscript to a native English speaker to improve the grammar, punctuation and sentences. There are a lot of grammatical inconsistency

Reviewer #3: 1. In the introduction, the sentences are too long; kindly consider shortening them by rewriting for better understanding.

2. Authors used terms like “alkali excitation, alkali-inspired, alkaline-stimulant” in some places, and in some other places it is alkali-activation; recommended to use the same term for the developing process.

3. “The material is a green geopolymer cementitious material that can replace cement” This sentence is repeated in the introduction.

4. In the introduction, the authors mentioned the main components of slag as silica and alumina, but the XRF results show that the main components are calcium oxide and silica oxide. Kindly check.

5. The introduction has to be rewritten by considering delivering ideas properly with short and clear sentences.

6. The novelty of the work is not clear. A lot of research work is available on the effect of activator dosage on binary blended alkali-activated concrete. Authors have to clearly state the novelty of the work.

7. Kindly check the unit of density mentioned in the paper.

8. The authors didn’t mention how the activator solution is prepared, the molarity, etc. and the percentage of alkali content is mentioned, but it is not clear about the percentage of what.

9. “Alkaline exciter: Hui-lime brand lime powder produced by Yixian Huangbaozhu Daily Chemical Products Co., Ltd, with CaO content of 93.45%”, which activator contains CaO content of 93.45%?

10. It is recommended to avoid long sentences in the discussion part for better understandability.

11. The authors mentioned that the quick setting of slag-based mortar is due to the “lower degree of polymerisation of the vitreous structure in the slag” Is this verified?

12. What about the pH of both activator solutions?

13. Ca is not soluble in water, so how does CaOH2 improve the dissolution of particles from fly ash? Authors must clearly explain the reaction mechanism.

14. The results and discussion part has to be strengthened with justification for the results, and compared with other literature available.

15. Kindly check the SEM images shown in Fig. 8, some mixes seem to be missing.

16. The title for the FT-IR study is provided as FIRT, kindly correct it.

17. The conclusion part should be crisp, and the outcome of the work should be delivered properly.

18. Include the images of materials used, mixing, casting, and testing of the cured specimen.

19. It is highly recommended to go through the paper and make the necessary corrections required to improve the quality of the research article.

**Do you want your identity to be public for this peer review?** For information about this choice, including consent withdrawal, please see our Privacy Policy

Reviewer #1: No

Reviewer #2: **Yes: ** Abiola Adebanjo

Reviewer #3: No

---

## [Author Response · Author response to Decision Letter 1]

7 Jul 2025

Dear Editor Kathirvel,

Thank you and the reviewers for your careful evaluation and valuable suggestions on our manuscript titled "[Your Paper Title]" (Manuscript ID: [Your Manuscript Number]).

We have thoroughly revised the manuscript according to the reviewers’ comments and have submitted the following materials via the submission system:

Response to Reviewers letter (labeled "Reviewers 1, Reviewers 2, Reviewers 3"): Addressed all points raised by the editor and reviewers point-by-point;

Revised manuscript with tracked changes (labeled "Revised Manuscript"): Clearly highlights all modifications;

Final clean version (labeled "Revised Manuscript -copy"): The complete revised manuscript without revision marks.

Yours Sincerely,

Pro. Aimin Gong

College of Hydraulic Engineering Yunnan Agricultural University

E-mail: 13708457658@163.com

 Reviewer1

Comments 1: The submission overall follows the lab instruction and is straightforward. It mimics the Chinese guidelines and thus based on other international known guidelines like ASTM its generalizing might become difficult. Overall, it fails to highlight gaps or biases in the literature, and provides uncharacterized research gaps leading to unjustified conclusions without meaningful insights or future directions.

Response 1: Thanks for your suggestion. This study strictly adhered to the National Standard of the People's Republic of China: "Method for testing cementitious sand strength (ISO method)" (GB/T 17671-2021), which equivalently adopts the international standard ISO method (International Organization for Standardization).The "Conclusions" section has been revised accordingly, with modifications highlighted in red. Furthermore, this section has been supplemented with the limitations of the present study and specific aspects warranting further investigations.

Comments 2: Most of the references are in Chinese which are not applicable for international readers

Response 2: Thanks for your suggestion. We agree with this comment and have supplemented the references in the revised manuscript. The revised content is indicated in the References section, highlighted in red.

Comments 3: Structurally disorganized. It doesn’t follow the IMRAD. It has figs with different parts while almost none of them either in the context or in the captions have been mentioned or discussed. For example, Fig 7, 15.

Response 3: Thank you for your comments. We have restructured the manuscript according to IMRAD format, which now includes not only the core sections (Introduction, Methods, Results, and Discussion) but also the title, author affiliations with complete addresses, abstract, keywords, and reference list. Additionally, detailed captions have been added to all figures, and relevant discussions have been substantially expanded. We appreciate your understanding of these essential revisions.

Comments 4: Sensitivity analysis as described in this work is DECLINED to be accepted

Response 4: We completely agree with your opinion and consent to remove the sensitivity analysis section from the manuscript.

Comments 5: Discussion on XRD Lack of convincing and documented Discussion: technical limitations/uncertainty analysis/feasibility study and possibility for large scale use/benefits to industry/solid comparison with other works to show the improvement/the impact of bias of the used data on the results/…

Response 5: Thank you for your relatively comprehensive discussion and suggestions regarding the XRD analysis. In response to the comments on the paper, this study, based on the phase analysis function of XRD, determines the types, contents, and structures of the various constituent phases in the material through qualitative and quantitative analysis of the crystal composition. The revised and supplemented content can be found in the red annotations on pages 17 to 19, specifically in section " XRD Analysis". Furthermore, we fully concur that the research on "the potential for large-scale application" is insufficient. Future research needs to delve deeper into the adaptability of alkali-activated geopolymer cementitious materials for different engineering application scenarios. While the study provides the optimal dosages for the two types of alkali activators, the reliability of these findings compared with other research also requires further investigation.

Comments 6: Substandard English/unreflective and general keywords/…

Response 6: We sincerely appreciate your valuable feedback on the linguistic quality and keyword selection of our manuscript. In response to your suggestions, we have implemented comprehensive revisions with detailed adjustments. Additionally, the Conclusions section has been augmented with recommendations for future research, as highlighted in red.

Comments 7: When we discuss setting time, procedure requires continuous updating (dynamic databases) while this work doesn’t provide a solution for uniformly normalizing dynamic databases via different factors. Search for new approaches via keywords like normalizing large scale sensor-based data with an automated method …

Response 7: Thank you for pointing this out. We fully concur with this comment. The time settings specifically prioritize controlling the initial and final setting times of the cementitious materials. In this study, the initial setting time was configured to be no earlier than 45 minutes, while the final setting time was set not to exceed 600 minutes. The tests were conducted in accordance with the Chinese National Standard "Test methods for water requirement of normal consistency, setting time and soundness of cement" (GB/T 1346-2011), utilizing a Vicat apparatus. Regarding novel measurement methodologies, such as sensor-based data acquisition systems, these represent an important direction for future research endeavors.

Comments 8: In terms of the strength (Fig 5, 6…) you may benefit from other works like https://link.springer.com/article/10.1007/s41062-016-0016-9...

Response 8: Thank you for your suggestions. Regarding the strength analysis, we have added the test values for compressive and flexural strength to the figures, as seen in Figures 5, 6, 7, and 8. Furthermore, we have carefully studied your article. The multivariate statistical method used for predicting the strength of marl has provided us with important insights for optimizing the correlation analysis between the microstructure and strength of alkali-activated cementitious materials. In future work, we will draw upon this method to delve deeper into exploring the quantitative relationship between the evolution process of hydration product phases and the ultimately achieved mechanical properties of the material.

Comments 9: Conclusion should be justified

Response 9: We sincerely appreciate your valuable feedback and have accordingly revised and supplemented the Conclusions section, with all modifications highlighted in red for your review.

Comments 10: Reference list should follow the journal guidelines and supplemented by DOI link

Response 10: We sincerely thank you for your thorough review and constructive suggestions. Regarding the issues of reference formatting and DOI supplementation, we will implement the following revisions: Strictly adhere to the journal's style requirements by unifying citation formats, supplementing DOI links for all references, providing a note indicating the absence of DOI for entries without one, and conducting a line-by-line verification of author names, publication years, punctuation marks, and other details to ensure full compliance with academic standards.

Reviewer 2:

Title

Comments 1: The title should be rephrased, fly ash and slag should appear in the title.

Response 1: We sincerely appreciate your suggestion! The manuscript title has been revised to: “Study on the Hydration Product Phase Evolution and Mortar Strength of Alkali-Activated Slag and Fly Ash Systems”.

Abstract

Comments 1:The abstract lacks quantity, include results obtained for compressive strength and flexural strength. Ensure that you contrast between results obtained for slag and fly ash.

Rewrite the abstract to improve readability。

Correct misspelt words

Response 1: Thank you for your comments. We have revised the abstract to add the results of compressive strength and flexural strength tests, clarify the difference in strength between slag and fly ash under alkali excitation, and modify the content as indicated in red on lines 26-29 of page 2. Spelling errors have also been corrected

Introduction

Comments 1: The justification for the study is not well established. Improve on this Rephrase sentence: is China's major contributor to emissions.

Response 1: Thank you for your suggestions. In the Introduction section, we have supplemented the background and significance of this study, adding a detailed discussion on alkali-activated geopolymer cementitious materials as a green alternative to cement. The revised content is marked in red annotations in lines 37–47 on pages 2–3 (Introduction section).Of course, as a substitute for cement, its engineering applications require further in-depth exploration and discussion. Additionally, the wording throughout the text has been revised.

Comments 2: se fullstop (.) before The Start as a new paragraph。

Response 2: We fully accept your suggestions and have revised the relevant sections accordingly.

Methods

Comments 1: Improve this paragraph to aid coherency. Also provide pictures of the materials used. It is important to provide the particle size distribution of the slag and the fly ash.

Response 1: We fully endorse this comment and have supplemented the manuscript with micrographs of the raw materials employed, as shown in Fig. 1. Additionally, the particle size distribution ranges of slag and fly ash have been incorporated into the materials description section. These revisions are highlighted in red on page 5, lines 99-102.

Comments 2: Provide detailed information about this procedures. Just as was done for compressive strength. Describe all the testing methods to a great detail in different subsections. XRD, FTIR, TAM, BSE

Response 2: We sincerely appreciate your thorough suggestions! The testing methods, including XRD, FTIR, TAM, and BSE, have been described in detail in the specimen preparation and testing sections. The change can be seen in red on pages 7 to 9, lines 126-174.

Results

Comments 1: What do you mean by change rule diagram.

The sentence is too long. Consider breaking the sentences to improve readability.

Response 1: We sincerely apologize for the inaccurate terminology " rule diagram" and have revised it. The change can be seen in red on pages 9 to 10, line 178-179. Additionally, overly lengthy sentences throughout the manuscript have been streamlined for enhanced clarity.

Comments 2: improve readability. Discuss relative to existing literature

Response 2: Thank you for your valuable feedback. We have carefully reviewed the manuscript and made revisions to improve readability and strengthen the discussion with existing literature.

Comments 3: The method for sensitivity analysis was not captured in the methods.

Response 3: Thank you very much for your thorough review of the manuscript and your valuable feedback. During the revision process, we decided to remove the section on sensitivity analysis from the original manuscript. This decision was made to preserve the core focus and overall integrity of the article.

Comments 4: Highlight the important minerals and show the observable changes using inscribed rectangles.

Response 4: We recognize the merit of highlighting mineral phases as proposed. However, the current analytical methodology yields data at a scale where precise mineral boundary delineation remains technically challenging.

Comments 5: FTIR analysis not FIRT Analysis

Introduce a shape here to call the reader to the difference in intensity.

Response 5: We sincerely appreciate you careful reading and valuable feedback. The term "FIRT" in the section title has been corrected to "FTIR". FTIR analysis has supplemented the difference in strength of slag and fly ash hydration systems under alkaline excitation. See pages 20 to 21, lines 378 to 395 marked in red.

Comments 6: Improve the discussion by making reference to recent literature. (Hydration characterization)

Response 6: We sincerely appreciate your suggestion to improve the discussion by incorporating recent literature on hydration characterization. In response, we have expanded the discussion in Section The revised content can be found in the red-highlighted text on page 24, lines 435–541.

Comments 7: Describe the method in detail under the method section. (Pore physical phase distribution and pore structure characteristics)

Response 7: Thank you for this valuable suggestion. As requested, we have added a detailed paragraph in the Methods section elaborating on the experimental procedures. highlighted in red on page 9, lines 164-167.

Comments 8: Consider rephrasing the sentence. Also improve this section with information from literature. What is the ideal pore area for geopolymer composites based on previous attempts by other researchers?

Response 8: We sincerely appreciate your suggestions. We have revised the sentences with problems in the paper's expression. Furthermore we analyzed the changes in pore number, maximum pore size, and total pore area. See pages 22 to 23, lines 400-417 marked in red, and point (3) in the "Conclusion".

Conclusion

Comments 1: Provide a brief information about the purpose of the study before itemising the conclusions. Also, include the limitations of this study after the conclusion. Furthermore, I will suggest you give the entire manuscript to a native English speaker to improve the grammar, punctuation and sentences. There are a lot of grammatical inconsistency.

Response 1: We fully endorse this recommendation. The opening paragraph of the Conclusion section now explicitly states the research objectives, while the concluding segment has been supplemented with the study's limitations. Additionally, comprehensive refinements have been made to grammar, punctuation, and sentence structure throughout.

Reviewer 3:

Comments 1: In the introduction, the sentences are too long; kindly consider shortening them by rewriting for better understanding.

Response 1: Thanks for your suggestion. The introduction section has been revised to shorten overly lengthy sentences, improve clarity, and enhance readability. The change can be seen in red on pages 2 to 5, in the introduction.

Comments 2: Authors used terms like “alkali excitation, alkali-inspired, alkaline-stimulant” in some places, and in some other places it is alkali-activation; recommended to use the same term for the developing process.

Response 2: Thank you for your careful review of our manuscript and for pointing out the inconsistency in terminology usage. We fully agree with your suggestion. Accordingly, we have adopted the standardized term "alkali-activation" and its related derivatives—such as "alkali-activated" and "alkali activator"—consistently throughout the manuscript to describe this process and relevant materials. This enhances the rigor and readability of the paper.

Comments 3: The material is a green geopolymer cementitious material that can replace cement” This sentence is repeated in the introduction.

Response 3: Thanks for your suggestion. We have removed the redundant sentences and improved the phrasing.

Comments 4: In the introduction, the authors mentioned the main components of slag as silica and alumina, but the XRF results show that the main components are calcium oxide and silica oxide. Kindly check.

Response 4: Thank you for pointing out this inconsistency. We apologize for this error and have corrected it. The corrected content is marked in red from lines 54 to 56 on page 3.

Comments 5: The introduction has to be rewritten by considering delivering ideas properly with short and clear sentences.

Response 5: We sincerely appreciate your valuable feedback. The introduction section has been thoroughly revised to enhance clarity and conciseness. Long sentences have been simplified, and key ideas are now presented in a more logical and reader-friend

---

## [Decision Letter · Decision Letter 1]

10 Oct 2025

Dear Dr. Gong,

Thank you for submitting your manuscript to PLOS ONE. After careful consideration, we feel that it has merit but does not fully meet PLOS ONE’s publication criteria as it currently stands. Therefore, we invite you to submit a revised version of the manuscript that addresses the points raised during the review process.

We look forward to receiving your revised manuscript.

Kind regards,

Parthiban Kathirvel

Academic Editor

PLOS ONE

Journal Requirements:

Reviewers' comments:

Reviewer's Responses to Questions

**Comments to the Author**

Reviewer #1: (No Response)

Reviewer #4: (No Response)

Reviewer #5: All comments have been addressed

2. Is the manuscript technically sound, and do the data support the conclusions?

Reviewer #1: Partly

Reviewer #4: No

Reviewer #5: Yes

3. Has the statistical analysis been performed appropriately and rigorously?

Reviewer #1: No

Reviewer #4: No

Reviewer #5: Yes

4. Have the authors made all data underlying the findings in their manuscript fully available?

Reviewer #1: No

Reviewer #4: No

Reviewer #5: Yes

5. Is the manuscript presented in an intelligible fashion and written in standard English?

Reviewer #1: No

Reviewer #4: No

Reviewer #5: Yes

Reviewer #1: The responses are appreciated. However, some technical justifications are required.

I actually was lost with the responses and place of modification corresponding to each comment and sub-comments. It is expected to see the responses for each individually

As you pretty well-mentioned ‘limitations of the present study and specific aspects warranting further investigations.’, the comments #5, 7 and 8 strictly emphasizes on technical limitations and you also notified them in your responses. Therefore, to enrich the literature, the draft in the section of technical limitation’ as a sub-category of the Discussion section should support and document these concerns. The suggestions were provided in the last round and you also have open hand to add new references to ensure the backup.

The problem regarding the feasibility study and capacity for working in large scale industrial applications also need to be added.

Please revisit the literal aspects to minimize the grammatical syntaxes.

Good Luck

Reviewer #4: The manuscript is poorly written and presented. I am unable to read the manuscript. All aspects of writing in English are completely ignored. Perhaps the authors can get assistance in English writing. One can not follow exactly how the experimental work was carried out.

Reviewer #5: After thorough review, the manuscript has successfully addressed all previous concerns and fully meets the scientific standards for publication. The paper is recommended for acceptance in its current form.

**Do you want your identity to be public for this peer review?** For information about this choice, including consent withdrawal, please see our Privacy Policy

Reviewer #1: No

Reviewer #4: No

Reviewer #5: No

---

## [Author Response · Author response to Decision Letter 2]

16 Oct 2025

of various activators with different geopolymers, along with the associated strength prediction models.

Comments 3: The problem regarding the feasibility study and capacity for working in large scale industrial applications also need to be added.

Response 3: We thank the reviewer for their valuable comments. We have now added the following statement to the Introduction section: "The utilisation of alkali activators to solidify industrial by-products and waste materials as cementitious alternatives not only effectively reduces the consumption of natural resources but also enables the harmless disposal and resource recovery of waste."

Comments 4: Please revisit the literal aspects to minimize the grammatical syntaxes.

Response 4: We thank the reviewer for this valuable comment. The manuscript has been carefully revised to address the grammatical and syntactic issues.

Comments 1: The manuscript is poorly written and presented. I am unable to read the manuscript. All aspects of writing in English are completely ignored. Perhaps the authors can get assistance in English writing. One can not follow exactly how the experimental work was carried out.

Response 1: We wish to express our gratitude for the reviewer's comments regarding the language and presentation of our manuscript. These suggestions have been duly incorporated to improve the clarity and readability of the paper.

---

## [Decision Letter · Decision Letter 2]

27 Oct 2025

Dear Dr. Gong,

Thank you for submitting your manuscript to PLOS ONE. After careful consideration, we feel that it has merit but does not fully meet PLOS ONE’s publication criteria as it currently stands. Therefore, we invite you to submit a revised version of the manuscript that addresses the points raised during the review process.

We look forward to receiving your revised manuscript.

Kind regards,

Parthiban Kathirvel

Academic Editor

PLOS ONE

Journal Requirements:

Reviewers' comments:

Reviewer's Responses to Questions

**Comments to the Author**

Reviewer #1: (No Response)

Reviewer #4: (No Response)

2. Is the manuscript technically sound, and do the data support the conclusions?

Reviewer #1: (No Response)

Reviewer #4: No

3. Has the statistical analysis been performed appropriately and rigorously?

Reviewer #1: (No Response)

Reviewer #4: No

4. Have the authors made all data underlying the findings in their manuscript fully available?

Reviewer #1: (No Response)

Reviewer #4: Yes

5. Is the manuscript presented in an intelligible fashion and written in standard English?

Reviewer #1: (No Response)

Reviewer #4: No

Reviewer #1: Dear colleagues

When we discuss technical limitations, they MUST be documented and thus references are required. This leads readers to find what really has been done and what is missing and what can be done for future studies. Therefore, with respect, your responses cannot be accepted in the current format. The reasons and suggestions have been provided in previous rounds. Furthermore, it is expected to see updates within the literature review process.

Conclusion is the conclusion. It is the last part of your puzzle and should show the summary of your sharp findings. Limitation MUST be moved to Discussion section.

Hope that the above mentioned clarifications help you with further concise responding.

Reviewer #4: The manuscript has adequate data that is presentable for publication. However, the manuscript still has been written in poor English. Methodology has improved but will still require to be worked on.

Results and discussion is poorly done. There is no scientific presentation of the results and discussion

**Do you want your identity to be public for this peer review?** For information about this choice, including consent withdrawal, please see our Privacy Policy

Reviewer #1: No

Reviewer #4: **Yes: ** Jackson Wachira Muthengia

---

## [Author Response · Author response to Decision Letter 3]

5 Nov 2025

Dear Editors and Reviewers,

We sincerely thank you for your letter and the valuable comments from the reviewers on our manuscript titled " Hydration Product Phase Evolution and Mortar Strength Development in Alkali-Activated Slag and Fly Ash Systems" (PONE-D-25-18472R2). These suggestions have been highly instructive in improving the quality of our paper. Based on the reviewers' feedback, I have thoroughly revised the manuscript, with all changes highlighted in red. We hope that the revised version meets the publication standards of your journal.

The reviewers' comments are presented in italics, with specific issues numbered. I have addressed each of these points individually, and my responses are provided in standard font and highlighted in red. I have made every effort to improve the manuscript and deeply appreciate the hard work of the editors and reviewers. The successful publication of this research is crucial for the completion of my degree application, and I sincerely hope that the revised manuscript will be accepted.

Thank you once again for your comments and suggestions.

Sincerely,

(Student's Name)

Supervised by: Professor Aimin Gong

College of Hydraulic Engineering, Yunnan Agricultural University

Email: 13708457658@163.com

---

## [Decision Letter · Decision Letter 3]

13 Nov 2025

Dear Dr. Gong,

Thank you for submitting your manuscript to PLOS ONE. After careful consideration, we feel that it has merit but does not fully meet PLOS ONE’s publication criteria as it currently stands. Therefore, we invite you to submit a revised version of the manuscript that addresses the points raised during the review process.

We look forward to receiving your revised manuscript.

Kind regards,

Parthiban Kathirvel

Academic Editor

PLOS ONE

Journal Requirements:

Reviewers' comments:

Reviewer's Responses to Questions

**Comments to the Author**

Reviewer #1: (No Response)

Reviewer #4: All comments have been addressed

2. Is the manuscript technically sound, and do the data support the conclusions?

Reviewer #1: Partly

Reviewer #4: Yes

3. Has the statistical analysis been performed appropriately and rigorously?

Reviewer #1: Yes

Reviewer #4: Yes

4. Have the authors made all data underlying the findings in their manuscript fully available?

Reviewer #1: Yes

Reviewer #4: Yes

5. Is the manuscript presented in an intelligible fashion and written in standard English?

Reviewer #1: No

Reviewer #4: Yes

Reviewer #1: After the following minor revision, the paper can be moved toward next step.

1. Reference format issues. For example, 11, 24, 25, , … [J/OL]!!!, …

2. Remove the Chiness characters

3. Writing Styles, some with all capital letters, some not, …

4. Some references like 7, 8, 14, 28, … don’t have any authors!!!!

5. Some references don’t have any DOI, like 5, …

6. Use DOI link (https://doi.org/...)

Reviewer #4: The manuscript is well done as per the requirements. Initially, the manuscript had a lot of grammatical errors. The manuscript is now well done.

**Do you want your identity to be public for this peer review?** For information about this choice, including consent withdrawal, please see our Privacy Policy

Reviewer #1: No

Reviewer #4: **Yes: ** Jackson Wachira Muthengia

---

## [Author Response · Author response to Decision Letter 4]

16 Nov 2025

Dear Reviewer,

We sincerely appreciate your valuable comments on our manuscript titled "Study on the Phase Evolution of Hydration Products and the Strength of Mortar in Alkali-Activated Solid Waste Materials" (PONE-D-25-18472R3). We are especially grateful for the insightful suggestions provided throughout multiple rounds of revision, which have deepened my understanding of academic rigor and significantly contributed to improving the quality of this paper. In response to your feedback, we have thoroughly revised the manuscript, with all changes highlighted in red.

The reviewer’s comments are presented in italics, and our responses are provided in regular font, also highlighted in red. We have made every effort to refine the content of the paper and sincerely thank you for your diligent work. We hope that the revised manuscript meets with your approval.

Thank you again for your comments and suggestions.

Yours Sincerely,

Pro. Aimin Gong

College of Hydraulic Engineering Yunnan Agricultural University

E-mail: 13708457658@163.com

Reviewer1

Comments : After the following minor revision, the paper can be moved toward next step.

1. Reference format issues. For example, 11, 24, 25, , … [J/OL]!!!, …

2. Remove the Chiness characters

3. Writing Styles, some with all capital letters, some not, …

4. Some references like 7, 8, 14, 28, … don’t have any authors!!!!

5. Some references don’t have any DOI, like 5, …

6. Use DOI link (https://doi.org/...)

Response : Thank you for your valuable comments and suggestions on our manuscript. We have carefully revised the paper according to your feedback and have highlighted all changes in red within the text. The specific modifications are as follows:

1.Reference Format: We have thoroughly checked and standardized the reference format. Irregular identifiers such as [J/OL] in entries (e.g., 11, 24, 25, etc.) have been corrected.

2.Removal of Chinese Characters: All non-essential Chinese characters have been completely removed from the manuscript.

3.Writing Styles: Inconsistencies in capitalization have been corrected to ensure consistency throughout the manuscript.

4.Missing Authors: Author information missing from reference entries (e.g., 7, 8, 14, 28, etc.) has been added.

5.Addition of DOIs: Digital Object Identifiers (DOIs) have been added for all applicable references (including entry 5, etc.).

6.DOI Links: The links for all references with DOIs have been uniformly formatted to the https://doi.org/... style.

---

## [Editor Report · Decision Letter 4]

18 Nov 2025

Hydration Product Phase Evolution and Mortar Strength Development in Alkali-Activated Slag and Fly Ash Systems

PONE-D-25-18472R4

Dear Dr. Gong,

We’re pleased to inform you that your manuscript has been judged scientifically suitable for publication and will be formally accepted for publication once it meets all outstanding technical requirements.

Kind regards,

Parthiban Kathirvel

Academic Editor

PLOS ONE

---

## [Editor Report · Acceptance letter]

PONE-D-25-18472R4

PLOS ONE

Dear Dr. Gong,

I'm pleased to inform you that your manuscript has been deemed suitable for publication in PLOS ONE. Congratulations! Your manuscript is now being handed over to our production team.

Kind regards,

on behalf of

Dr. Parthiban Kathirvel

Academic Editor

PLOS ONE